# Characterisation of tumour microenvironment remodelling following oncogene inhibition in preclinical studies with imaging mass cytometry

Febe van Maldegem [1,7,8✉], Karishma Valand [1,8], Megan Cole [1], Harshil Patel [2], Mihaela Angelova [3], Sareena Rana[1,4], Emma Colliver [3], Katey Enfield [3], Nourdine Bah[2], Gavin Kelly [2], Victoria Siu Kwan Tsang [1], Edurne Mugarza[1], Christopher Moore[1], Philip Hobson[5], Dina Levi[5], Miriam Molina-Arcas[1], Charles Swanton [3,6] & Julian Downward [1,4✉]

Mouse models are critical in pre-clinical studies of cancer therapy, allowing dissection of mechanisms through chemical and genetic manipulations that are not feasible in the clinical setting. In studies of the tumour microenvironment (TME), multiplexed imaging methods can provide a rich source of information. However, the application of such technologies in mouse tissues is still in its infancy. Here we present a workflow for studying the TME using imaging mass cytometry with a panel of 27 antibodies on frozen mouse tissues. We optimise and validate image segmentation strategies and automate the process in a Nextflow-based pipeline (imcyto) that is scalable and portable, allowing for parallelised segmentation of large multi-image datasets. With these methods we interrogate the remodelling of the TME induced by a KRAS G12C inhibitor in an immune competent mouse orthotopic lung cancer model, highlighting the infiltration and activation of antigen presenting cells and effector cells.

[1] Oncogene Biology Laboratory, The Francis Crick Institute, 1 Midland Road, London NW1 1AT, UK. [2] Bioinformatics and Biostatistics Science Technology Platform, The Francis Crick Institute, 1 Midland Road, London NW1 1AT, UK. [3] Cancer Evolution and Genome Instability Laboratory, The Francis Crick Institute, 1 Midland Road, London NW1 1AT, UK. [4] Lung Cancer Group, Division of Molecular Pathology, Institute of Cancer Research, 237 Fulham Road, London SW3 6JB, UK. [5] Flow Cytometry Science Technology Platform, The Francis Crick Institute, 1 Midland Road, London NW1 1AT, UK. [6] Cancer Research UK Lung Cancer Centre of Excellence, University College London Cancer Institute, Paul O'Gorman Building, 72 Huntley Street, London WC1E 6DD, UK. [7] Present address: Department of Molecular Cell Biology and Immunology, Amsterdam UMC, Vrije Universiteit Amsterdam, De Boelelaan 1108, 1081HZ Amsterdam, The Netherlands. [8] These authors contributed equally: Febe van Maldegem, Karishma Valand. ✉email: f.vanmaldegem@amsterdamumc.nl; julian.downward@crick.ac.uk

The advent of successful immune checkpoint inhibitors has revolutionised the treatment of cancer in recent years. However, a large proportion of patients exhibit intrinsic or acquired resistance, and good prognostic markers for response are lacking. The TME is thought to play a key role in mediating immune evasion. Therefore, it will be crucial to enhance our knowledge about the cells infiltrating the tumour and their spatial context. Manipulating the TME to revert immune suppression has the potential to significantly enhance the efficacy of immunotherapies. Mouse preclinical cancer models provide an excellent platform to study such interventions aimed at the TME in a controlled manner.

The use of multiplex techniques such as single cell mass cytometry (CyTOF) and single cell RNA sequencing, it has become apparent that tumours are infiltrated with a diverse spectrum of immune cells, often with different phenotypes from their normally homoeostatic counterparts[1–4]. Unfortunately, the digestion of the tissue that is required to perform such analysis destroys the spatial context of the TME. Immunofluorescence (IF) and immunohistochemistry can provide spatial localisation data, but the number of markers that can be used simultaneously is limited by the spectral overlap of fluorophores and chromogens. Thus, highly multiplexed imaging technologies, such as imaging mass cytometry (IMC) that is based on unique atomic mass are very attractive, permitting in depth characterisation of the TME with a metal-conjugated antibody panel of up to 40 markers while retaining the spatial context[5,6].

While products and methods for conducting IMC studies on patient samples are becoming well established, publications using IMC in the mouse are scarce and use limited antibody complexity without subsequent quantification of the images[7,8]. Here we present a complete IMC workflow, including a validated 27-marker antibody panel, automated and optimised image segmentation using our imcyto pipeline and showcase various spatial analyses. We applied these methods to study the effects of MRTX1257, a mutant-specific inhibitor of the KRAS G12C oncoprotein, on the TME of an immunotherapy refractory KRAS G12C mutant lung cancer. The targeted inhibition of oncogenic KRAS signalling using the recently developed mutant-specific KRAS G12C covalent inhibitors has shown very promising efficacy in phase 1 and phase 2 clinical trials and was recently approved for locally advanced or metastatic KRAS G12C mutant lung cancer[9–11]. However, it is expected that long-term therapeutic responses will be limited by the acquisition of drug resistance[12–14], and therefore combination therapies are under intense investigation. Two recent studies reported enhanced survival when combining the KRAS G12C inhibitors with anti-PD-1 immune checkpoint blockade in an immunotherapy sensitive syngeneic KRAS G12C mutant CT26 colon carcinoma subcutaneous tumour model[15,16]. This prompted us to investigate the effects of tumour-specific KRAS inhibition on the TME in the context of a preclinical model of lung cancer, the 3LL ΔNRAS cell line, a KRAS G12C mutant and NRAS-knockout Lewis lung carcinoma derivative that we have previously shown to be sensitive to KRAS G12C inhibition[17]. The Lewis lung carcinoma is considered to be highly refractory to immune interventions[18] and has an immune cold or immune excluded phenotype.

Here we present a workflow to enable the spatial and phenotypic characterisation of the tumour microenvironment in preclinical mouse models by IMC, including a mouse antibody panel, a cell type-optimised segmentation strategy and a scalable and portable pipeline for automated segmentation of large datasets. Subsequently, we employed a collection of analysis methods, such as clustering and dimensionality reduction to characterise the cells in the tissues. We integrated the spatial information at the tissue level using the domain classification obtained from our segmentation pipeline, while cell–cell interactions were explored using the neighbouRhood analysis developed by the Bodenmiller group[19], as well as approaches that assess distances between cell types, exploring different aspects of the spatial relationships within the TME. Using this methodology we gained detailed quantitative insight into the phenotypes and spatial relationships of immune cells and stromal compartments of the mouse lung TME, and how KRAS G12C inhibition promotes remodelling into a more immune activated state.

## Results

**Validated antibody panel for IMC on mouse tissues.** Due to the limited availability of well-validated antibodies for use on mouse formalin-fixed and paraffin-embedded (FFPE) tissues, especially with respect to immunology markers, we designed and validated a 27-antibody panel for use on frozen tissue sections. Example composite images are shown in Fig. 1a and Supplementary Fig. 1.

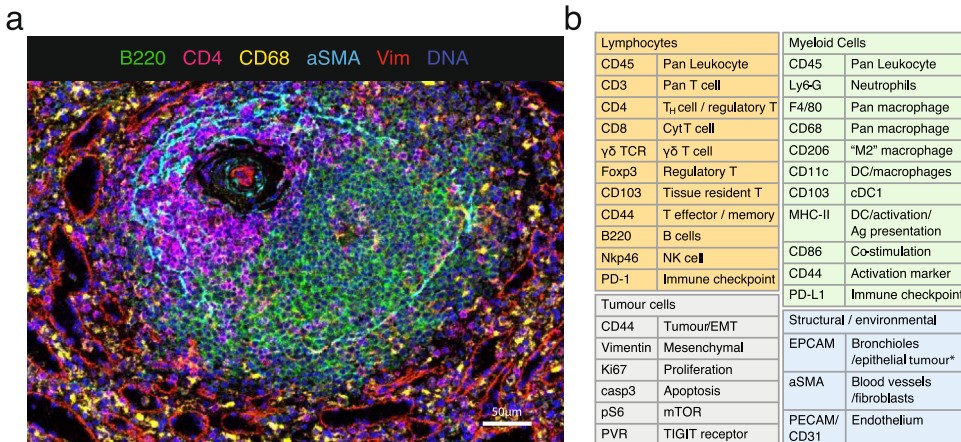

**Fig. 1 A 27 multiplex antibody panel to characterise the TME in mouse frozen tissues. a** Validation of antibodies in IMC using a tissue with known staining patterns, an example shown here is a follicle in the spleen; Ir191/193 (blue); B220 (green); CD4 (magenta), CD68 (yellow), αSMA (cyan), vimentin (red). Representative image of two independently stained spleen tissues. **b** Panel of 27 antibodies that identify multiple cell types from lymphoid, myeloid, tumour and stromal compartments, as well as markers of activation and proliferation status. Detailed information on the antibody clones and isotope conjugations can be found in Supplementary Tables 1 and 2. DC dendritic cells, NK natural killer, EMT epithelial-to-mesenchymal transition, Ag antigen.

With this panel, we are able to distinguish many different immune cell types that are thought to play a role within the TME, such as lymphocytes and various subsets of myeloid cells. In addition, this panel includes markers to visualise the context of the tissue architecture, e.g. endothelium and fibroblasts, as well as phenotypic markers that describe maturation and activation state of both tumour and immune cells (Fig. 1b and Supplementary Table 2). Several metals were kept free for insertion of additional markers to customise this panel, with some slots in the most sensitive range of the detector, suitable for dimmer markers such as additional checkpoint molecules.

**Optimisation of single cell segmentation using cell type-specific settings.** Quantification of IMC data requires image segmentation to extract expression data at the single cell level. We sought a software solution that would be able to scale sufficiently and therefore chose to work with open-source packages Cell-Profiler and Ilastik, as previously described by the Bodenmiller lab[20]. Cell Image Analysis Software CellProfiler was used for pre-processing and subsequent object identification and segmentation. Ilastik provides interactive pixel classification to generate staining probability maps that can be used to even out variations in staining intensities at the local level and between image sets, providing a more robust basis for segmentation. All of the published segmentation strategies for IMC were originally optimised and validated on human FFPE tissues. Therefore, we first questioned what method would be most appropriate for use on frozen mouse tumours. In most instances, cell segmentation starts with the segmentation of the nuclei, which can subsequently be used as seeds to expand into cell objects. Various methods have been described to subsequently define the cell border: by expanding the nuclear perimeter by a fixed distance, such as three pixels, by a watershed method onto a membrane staining[6], or by propagation onto a probability map of membranes generated by pixel classification with machine learning[20,21]. We started with testing two of these strategies, the three-pixel expansion and the propagation method onto a membrane probability map (Fig. 2a), using a tumour from the 3LL Lewis lung carcinoma stained with our antibody panel. From visual inspection of the cell masks, we observed several recurring problems such as the failure to capture membrane staining of large cells, and the inclusion of signals from neighbouring cells into the smaller cells such as lymphocytes. In particular, the inclusion of signals from neighbouring cells (spatial "spillover") led to increased noise in the data. In addition, we observed that cells with a spindle shape, such as alpha-smooth muscle actin (αSMA) expressing fibroblasts, were captured very poorly. Figure 2b shows examples of these segmentation errors resulting from three-pixel expansion and propagation pipelines in studying F4/80+ macrophages, CD4+ T cells and aSMA+ fibroblasts.

We explored modifications of these segmentation strategies with the aim to improve the quality of the data. To improve the signal-to-noise ratio in the pixel expansion segmentation we minimised the radius by which the cells were expanded from the nucleus. A one-pixel expansion segmentation led to reduced contact between cell boundaries and therefore less signal bleed between neighbouring cells (Fig. 2b). This was reflected in the improved signal capture ("enrichment") per cell when considering the amount of signal capture in the brightest 500 cells for a particular marker, relative to the rest of the cells in the image (Fig. 2c), as well as the improved signal/noise within individual cells (Fig. 2d).

These three strategies used only one set of parameters to identify all nuclei, the seeds of each cell object. However, in complex tissues such as the TME, nuclei differ widely in size and shape. Setting a size threshold for an average tumour cell may easily exclude the smaller T cells, while stringent 'declumping' settings can unnecessarily split up larger multilobed nuclei of tumour cells and macrophages. Also, the general absence of a clearly identifiable nucleus associated with the αSMA signal was a likely cause for the poor segmentation of fibroblasts. Therefore, we modified the propagation method, to sequentially segment the different cell types using settings that are optimal for those cells. Each of the 'segmentation layers' is based on its own probability map and can use different settings for nuclear identification. Figure 2a includes a graphic explaining the different layers of the sequential segmentation pipeline. By subtracting the mask of the cells identified at each step from the primary object layer (the nuclei), we can prioritise the cell types in the order at which they are segmented. The prioritised segmentation of the smaller lymphocytes improved their signal enrichment and signal-to-noise ratios, compared to the standard propagation method. A significant improvement was also seen for the capture of the αSMA signal into segmented cells (Fig. 2b–d).

As further validation of segmentation quality, we measured how well each method was able to detect challenging cell types, such as small lymphocytes, irregularly shaped macrophages and the before mentioned fibroblasts. We manually annotated a set of CD4+ and CD8+ T cells, CD103+ dendritic cells (DCs), F4/80+ macrophages and αSMA+ fibroblasts in the image. We counted how many of the highest expressing cells identified by the different segmentation strategies overlapped with these manually annotated cells. There was an improvement in detection of T cells, macrophages and fibroblasts in the sequential segmentation dataset, as shown in the bar graph manual annotation in Fig. 2e. As a negative control we also looked at the identification of CD103+ DCs, which was not prioritised in the sequential segmentation pipeline using cell type-specific markers, but instead was segmented at step 9 along with the bulk of the remaining cells using one-pixel expansion. These cells were detected to a similar extent in all four segmentation pipelines.

An added advantage of using pixel classification with Ilastik was that we could also use marker combinations to identify larger tissue structures, including tumour, normal adjacent tissue and the interface between those two. This domain segmentation added a layer of spatial context, allowing for quantification of markers and cell types within the different tissue compartments (Fig. 2f).

**imcyto: a Nextflow-based automated pipeline for segmentation on a high-performance computing cluster.** For the automated processing of our complex sequential segmentation method on large datasets, we developed a Nextflow-based segmentation pipeline, optimised to run on a high-performance computing cluster. Nextflow is based on the concept that software does not need to be installed on a server for it to be executed. Using Docker or Singularity containers, software packages are run as virtual images and are thereby portable, reproducible and platform-independent. The resulting pipelines can be parallelised to make the process scalable across large datasets[22]. For automation of our workflow, we combined IMCtools, Ilastik and CellProfiler into our Nextflow-based "imcyto" pipeline (Supplementary Fig. 2). The programmes take in custom settings provided in the form of user-generated Ilastik and CellProfiler project files as plugins, thereby making the pipeline adaptable to widely varying segmentation methods. As such, all of the four segmentation strategies discussed above can be run using this pipeline. Segmentation using the sequential method on our dataset consisting of 12 images ranging from 429 Mb up to 3.35 Gb, was completed in less than 45 min on the high-performance cluster at The Francis Crick Institute.

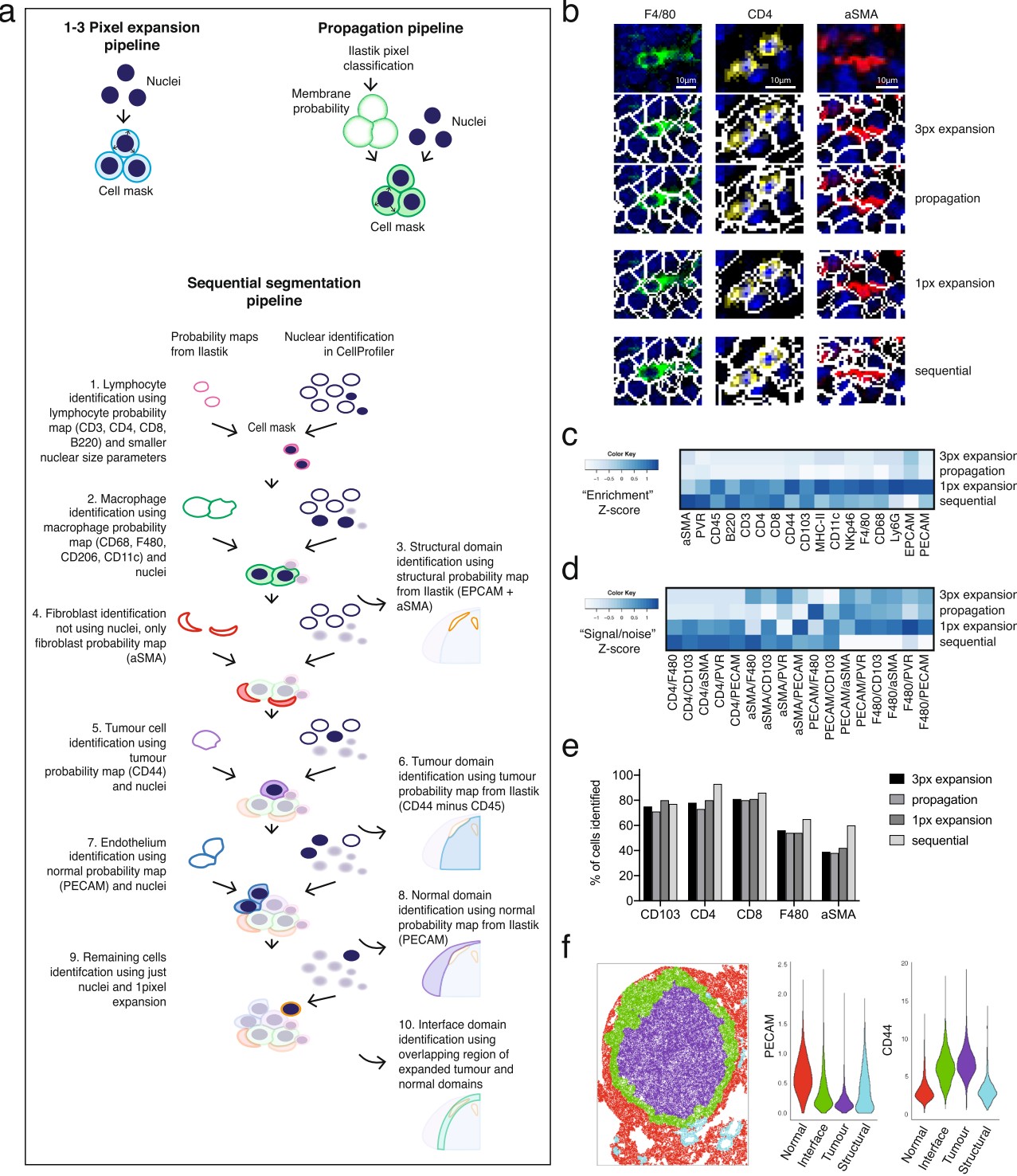

**Targeted KRAS G12C inhibition in a preclinical model of NSCLC.** Eight mice, harbouring orthotopic tumours of the 3LL ΔNRAS cell line, were treated for 7 days with the KRAS G12C inhibitor MRTX1257 ($n = 3$) or vehicle ($n = 5$). Treatment with MRTX1257 very markedly limited tumour growth compared to vehicle-treated mice, although significant tumour regression was not observed (Supplementary Fig. 3). From three mice of each group, the lungs were harvested, processed and stained for IMC. A total of 12 regions of interest (ROIs) ranging from 1–9 mm² were selected for downstream analysis with IMC; six tumours for each treatment group and where possible including adjacent normal tissue.

**IMC of lung tumours treated with MRTX1257 KRAS G12C inhibitor.** At first inspection, the images revealed recurrent patterns in the cellular arrangement of the tissues (Fig. 3a). Vehicle-treated tumours showed a tissue architecture in which CD68+ macrophages accumulated along the edge of the tumour, while F4/80+ and CD206+ macrophages seemed more intermixed with tumour cells. Aggregates of neutrophils were often observed associated with necrotic regions within the tumour and effector cells such as CD4+ and CD8+ T cells were mostly excluded from the tumour area. Upon treatment with MRTX1257, the tissue organisation became more diffuse, the macrophages were more

**Fig. 2 Comparing four segmentation strategies. a** Graphic describing segmentation strategies performed in CellProfiler. Identified nuclei are expanded on by a select number of pixels to create a cell mask with the pixel expansion strategy. The propagation strategy uses both identified nuclei and a membrane probability map generated by pixel classification in Ilastik to create a cell mask, using the propagation thresholding parameter in CellProfiler. In the sequential segmentation pipeline, each step uses custom size threshold settings for nucleus detection, as well as cell type-specific markers to generate the probability maps for membranes in Ilastik. A propagation step as described in (**a**) is subsequently used to find cell boundaries in steps 1, 2, 4, 5 and 7. At every level, the identified objects are subtracted from the total remaining nuclei. Any remaining cells at step 9 are segmented using one-pixel expansion. Steps 3, 6, 8 and 10 describe the segmentation of tissue domains, based on probability maps in Ilastik. **b** Representative false colour images of F4/80 (green), CD4 (yellow) and aSMA (red) markers merged with a nuclear stain (blue) and cropped (top row). Cell mask outlines (white) overlaid onto these markers were generated by either three-pixel expansion strategy (second row) or propagation strategy (third row), one-pixel expansion strategy (fourth row) or sequential segmentation (bottom panel). Image processing for visualisation: outliers were removed and a median filter of 0.5-pixel radius was applied in Fiji. **c** Heatmap showing enrichment of markers as a result of each segmentation strategy. Enrichment was defined as the relative expression in the top 500 cells of the markers as listed on the x-axis, compared to the expression of those same markers in the rest of the cells in the dataset. **d** Heatmap depicting the relative amount of noise in each segmentation strategy by looking at the relative expression of the key identifying marker, compared with markers that would not be expressed on the same cell, but may be found in its direct proximity within the tissue and thus would be a sign of signal bleed from adjacent cells ("signal/noise"). **e** Percentage of cells from the manually annotated dataset that were matched with cells in the top 500 for each marker, compared between segmentation strategies. Size of annotated datasets: CD103$^+$ DCs, $n = 65$; CD4$^+$ T cells, $n = 92$; CD8$^+$ T cells, $n = 70$; F4/80$^+$ macrophages, $n = 108$; αSMA$^+$ fibroblasts, $n = 130$. **f** Graphic showing the result of the domain segmentation as part of the sequential segmentation method, red = normal tissue, purple = tumour, green = interface, cyan = structural domain. Violin plot: Quantification of two key markers PECAM and CD44 used as the basis for the domain segmentation. px pixel.

apparently abundant within the tumour, the neutrophil presence diminished, and most dramatically, the T cells infiltrated the tumour bed.

**Segmentation and clustering identified 13 cell types in the TME.** The 12 ROIs were segmented using the sequential segmentation pipeline described above, giving a dataset of 282,837 cells, including mean intensities for all 27 markers, x and y coordinates and domain assignments. Unsupervised clustering by Phenograph[23], followed by supervised splitting of clusters and manual annotation of cell types based on marker expression patterns, resulted in a total of 35 clusters annotated into 13 cell types (Fig. 3b, Supplementary Fig. 4a and Supplementary Table 3). Approximately 37% of cells in the whole dataset were tumour cells and a similar proportion was taken up by myeloid cells, predominantly macrophages. Fibroblasts and lymphocytes represented a much smaller proportion, 2.2 and 5.8% respectively (Supplementary Fig. 4b). Principal component analysis (PCA) on the mean intensities of all markers per cell showed separation of the images by treatment along the principal component axes (Supplementary Fig. 4c). In particular, the macrophage and fibroblast compartments were expanded by the MRTX1257 treatment, while the proportion of neutrophils appeared reduced (Supplementary Fig. 4b, d, e). The domain distribution of the cell types showed large shifts in a spatial organisation as a response to the treatment (Fig. 3c and Supplementary Fig. 4e, f). Many cell types had an increased presence in the tumour domain in the MRTX1257 treated samples, such as lymphocytes, which we had already observed in the raw images (Fig. 3a), but also fibroblasts, DCs and endothelium, with exception of the neutrophils that were no longer abundantly present in the tumour. To interactively explore the movement of the cell types between the tissue domains as a result of the treatment, we made use of a visualisation tool that sets the three main tissue domains 'normal', 'tumour' and 'interface' as x, y and z dimensions in a 3D plot. By connecting the averages of the two treatment groups this visualised the magnitude of the shift in spatial distribution (Supplementary Fig. 4g and Supplementary Note 1). While the MRTX1257 treatment achieved good inhibition of tumour growth over the treatment period of 7 days, there was very little actual tumour regression observed. We wondered whether some of this could be attributed to the increased influx of immune cells, as has been described in the form of pseudoprogression or stable disease in response to immune checkpoint inhibitors[24,25].

Comparing the relative cellularity of the main cell types between the treatments Fig. 3d confirmed that the proportion of tumour cells in the tumour domain was reduced from 66% in the vehicle-treated dataset to 40% in the MRTX1257 treatment group, while many immune subsets, particularly the macrophage compartment, had expanded significantly, supporting the idea that the amount of tumour regression may have been underestimated in the tumour volume measurements by microCT.

**Macrophages subsets exhibit different tissue localisation and treatment responses.** Canon et al. previously reported changes in macrophages in response to the KRAS G12C inhibition[15]. This prompted us to investigate these cell types in more detail. Uniform Manifold Approximation and Projection for Dimension Reduction (uMAP), which better preserves the global structure in the data than tSNE[26], confirmed the relatedness of many of the macrophage Phenograph clusters (Fig. 4a). The macrophage population was separated out into two major macrophage subtypes that not only differed by phenotype, but also by their distribution across the tissues and how they responded to the treatment. One subset, which we called Type 1 macrophages, were mainly represented by cluster 11 and characterised by high CD68 and CD11c expression, PD-L1 expression and ribosomal protein S6 phosphorylation (pS6) (Fig. 4a, b). The expression of CD11c and CD68 is similar to lung resident alveolar macrophages but they lack the characteristic CD206 expression[27–29], and their accumulation at the tumour edge (interface domain) and some expression of pS6 suggests these might be reactive M1 polarised macrophages[30,31]. This macrophage subset remained fairly stable between the two treatment groups, with respect to phenotype and localisation (Fig. 4c). The second subset of macrophages, Type 2, represented by clusters 5 A, 8 and 26, was found almost exclusively within the tumour domain, and characterised by F4/80 and some CD206 expression (Fig. 4b). The expression of CD206 on this subset is striking, as CD206 is usually expressed on normal alveolar macrophages, but also considered a marker for immune suppressive M2 tumour-associated macrophages. This macrophage population expanded significantly upon KRAS inhibition. Furthermore, their phenotype was dramatically different upon treatment, with higher expression of F4/80, CD206 and most notably MHC-II and CD86, indicating an increased presence of mature antigen-presenting cells (Fig. 4a).

Using CellProfiler's 'Object relationships' output from our imcyto pipeline, we applied the Bodenmiller neighbouRhood

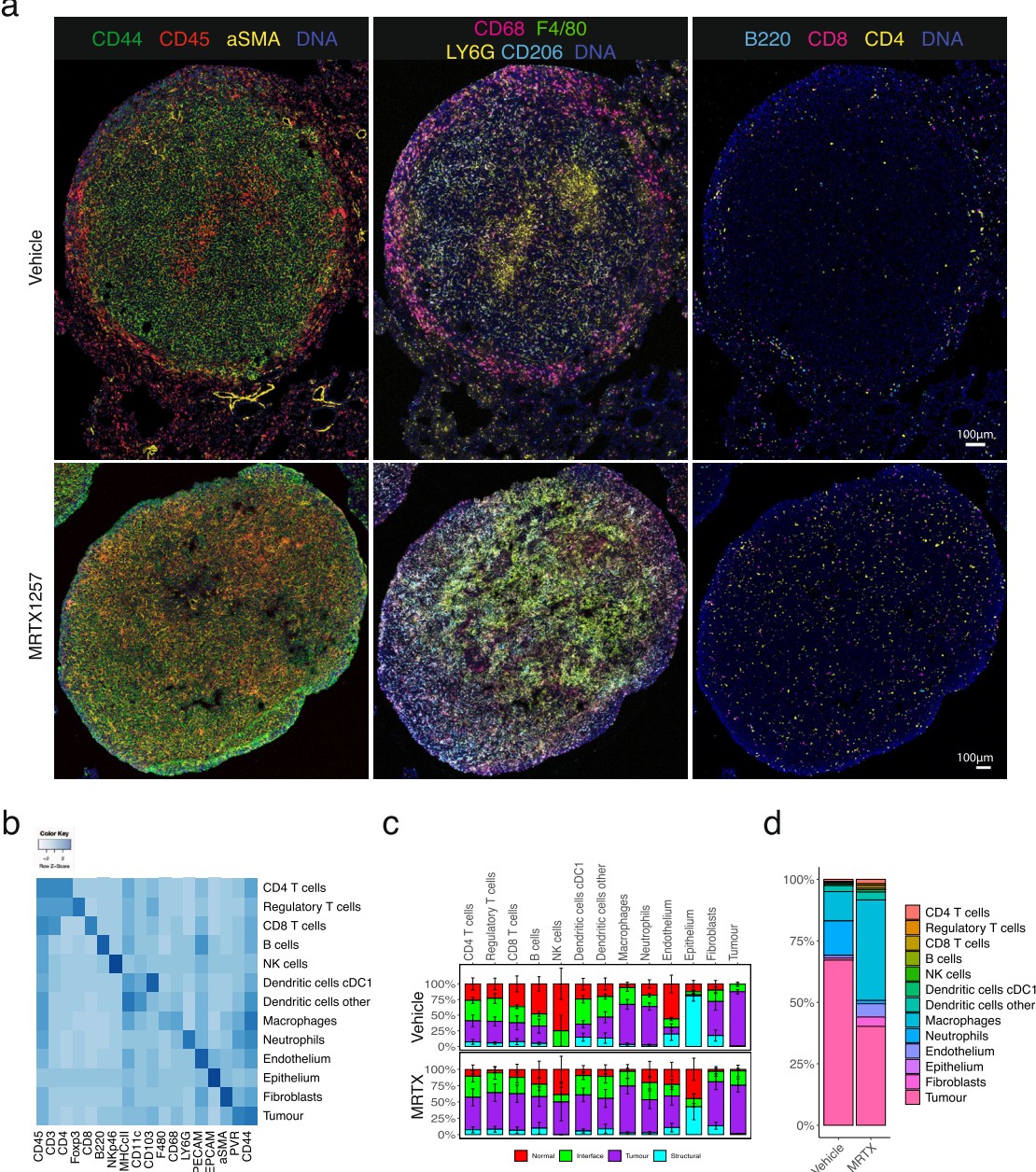

**Fig. 3 Characterisation and spatial distribution of cell types in Lewis lung carcinoma model. a** Two representative 3LL lung tumours, either treated with vehicle or MTRX1257 for 7 days ($n = 6$ for both groups). Overall tissue organisation changes upon KRAS inhibition, with increased CD45+ leucocytes and αSMA+ fibroblasts in the tumour domain (left), increased expression of macrophage markers such as F4/80 and CD206 in the tumour (middle), and more T cells infiltrating the tumour bed (right). For visualisation purposes, the images were processed in Fiji with a median filter (radius 0.5). To enhance visualisation of the small lymphocytes, the channels for CD3, CD4, CD8 and B220 were filtered using a band pass Fast Fourier Transformation. **b** Heatmap showing the expression of 18 lineage markers within the 13 identified cell types. The heatmap was scaled by row to emphasise the key markers per cluster. **c** Relative distribution of cell types within the tissue domains, separated by the two treatments. It shows how the fibroblasts, lymphocytes, DCs and macrophages have increased presence in the tumour domain, while neutrophils have been reduced. Error bars indicate the standard error of the mean over the ROIs ($n = 6$ for each treatment group). For full statistics using mixed-effects model, see Supplementary Fig. 4. **d** Proportions of cell types within the tumour domain, separated by treatment. Abbreviations MRTX MRTX1257.

analysis[19] on the two macrophage subtypes to look for cell types that would be enriched or depleted in their direct neighbourhood. Rather than looking at the number of images for which a spatial relationship was found significant, which is the default output of this analysis, we looked at the log2 fold change in neighbourhood enrichment compared to the permutated data for each ROI, to reflect the magnitude of enrichment or depletion of cell types in each other's neighbourhoods (Supplementary Fig. 5a). This

revealed some interesting relationships, such as the consistent coexistence of Type 1 macrophages with DCs and to a lesser extent CD4+ T cells (Fig. 4d) in particular in the interface domain (Supplementary Fig. 5b, c), while they were being rarely found in close contact with tumour cells. Indeed, visualising these cell types showed that Type 1 macrophages are found often in close proximity to CD4+ T cells and DCs, particularly at the edge of the tumour (Fig. 4e). This is also true of the minority of intra-

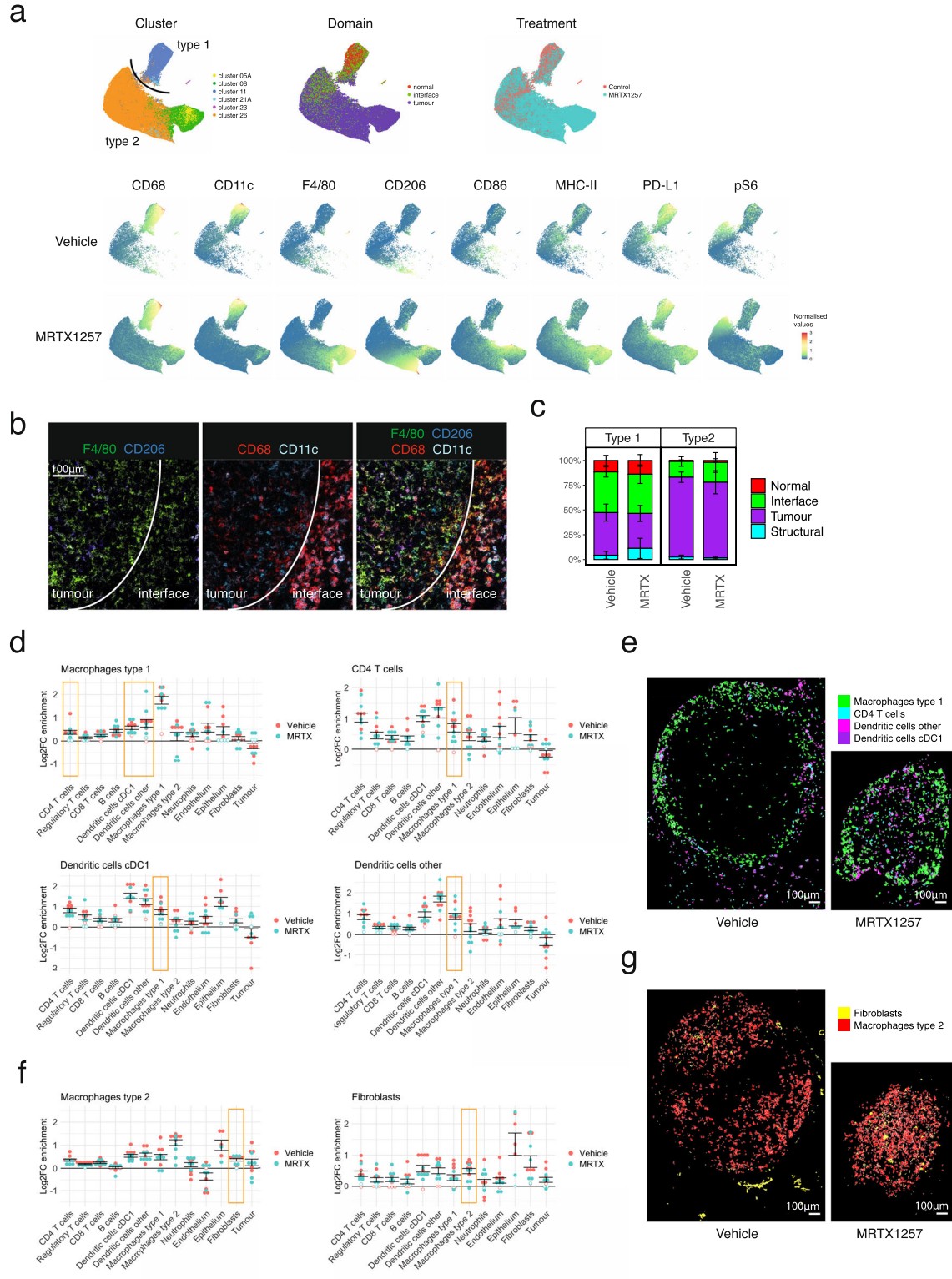

tumoural Type 1 macrophages, which maintain neighbourhoods with dendritic cells. Type 2 macrophages interact with DCs to a similar extent as the Type 1 macrophages, and in addition show an interesting spatial interaction with fibroblasts, where in particular the fibroblasts in the tumour domain and at the interface were found close to Type 2 macrophages (Fig. 4f, g and Supplementary Fig. 5b, c). This is in line with reports that cancer-associated fibroblasts can recruit monocytes and differentiate them to M2-like tumour-associated macrophages[32].

**T cells infiltrate the tumours in response to MRTX1257.** Lymphocytes are thought to be the most important effector in the antitumour immune response. In the work from Canon et al.[15], it was noted that KRAS inhibition led to an increase in T cells in the tissue, with greater expression of the activation and exhaustion marker PD-1. In the images of our mouse lung tumours we observed a modest but not significant increase in T cell numbers as a consequence of the treatment, but more strikingly they were now able to migrate into the tumour

**Fig. 4 Impact of tumour-specific KRAS G12C inhibition on phenotype and spatial characteristics of macrophages. a** uMAP of macrophage clusters separates into two distinct populations (top left). Repeated sampling and uMAP dimensionality reduction gave a similar distribution of the data ($n = 3$), representative plots shown. uMAP of macrophages, coloured by domain (top middle), treatment (top right), and the expression of individual markers that were selected for the generation of these uMAPs in the bottom two rows, data separated by treatment group. Type 1 macrophages, largely made up of cluster 11, are characterised by high CD68 CD11c expression, as well as PD-L1 and some pS6 positivity, and mainly reside at the tumour edge (normal and interface domain) and do not change much upon treatment with MRTX1257. The Type 2 macrophages, a merge of clusters 05A, 08 and 26, is characterised by F4/80 and CD206 expression. This population is much increased in size upon treatment, predominates the tumour domain and shows upregulation of maturation markers F4/80, CD206, MHC-II, CD86 and PD-L1 with treatment. **b** Crop of a vehicle-treated tumour to illustrate the difference in markers expression between the CD68+ CD11c+ Type 1 macrophages at the interface and F4/80hi and CD206+ Type 2 macrophages in the tumour. For visualisation purposes, the images were processed in Fiji with a median filter (radius 0.5). **c** Tissue distribution for both macrophages differs little between treatments. Stacked bar graph showing the distribution of the two macrophage types within the tissue domains. Type 1 macrophages are found mostly in the normal and interface domain. Type 2 macrophages are found almost exclusively in the tumour domain. Error bars indicate the standard error of means for the averages of proportions per ROI. Linear mixed-effect modelling based statistics for this data are depicted in Supplementary Fig. 4f. **d** Log2 fold changes in enrichment from the neighbouRhood[19] analysis for Type 1 macrophages, CD4+ T cells and dendritic cells. Filled circles represent images for which the enrichment was statistically significant compared to randomisation of all events in the image as calculated within the neighbouRhood package, while open circles indicate non-significance. Values above zero indicate enrichment of a cell type in the neighbourhood of the macrophages, below zero means depletion. Error bars indicate the standard error of means between the ROIs. **e** Visualisation of cell outlines of CD4+ T cells and dendritic cells, to demonstrate the frequent occurrence in the close proximity of Type 1 macrophages in two representative ROIs ($n = 12$). **f** Log2 fold change enrichment for Type 1 macrophages and fibroblasts as in **d**. **g** Visualisation of cell outlines of fibroblasts, to demonstrate the frequent occurrence in the close proximity of Type 2 macrophages in two representative ROIs ($n = 12$). MRTX MRTX1257.

domain (Fig. 3a, c and Supplementary Fig. 6a–c). uMAP visualisation of the T cell clusters in our dataset yielded a good separation of CD4+, CD8+ and regulatory T cells (Foxp3+) and even a few rare gamma-delta T cells (Fig. 5a). Similar to what has been previously reported, PD-1 expression on CD8+ T, CD4+ T and regulatory T cells was markedly increased upon KRAS inhibition (Fig. 5b). More interestingly, PD-1 expression was mostly confined to T cells within the tumour area, and not detected in the adjacent normal tissue. This suggests that there is either selective recruitment of activated T cells into the tumour domain or, perhaps more likely, a localised induction of PD-1 when T cells come into contact with the TME. Insight into which cell types come into close contact with the T cells could shed some light on this matter. The neighbouRhood analysis looks at local enrichment and depletion of cells in the direct surroundings of a cell of interest, but the correction towards a permuted dataset can lead to a relative under-representation of spatial interactions with abundant cell types. To quantify in a more absolute way how the neighbourhood of the T cells changes with treatment, we decided to look at distances between cell types and the frequency of those interactions. We calculated the distance of each cell to the nearest CD4+, CD8+ or regulatory T cell using Pythagoras's theorem, which gives the spectrum of proximities between cell types and works across the whole image. This approach showed that some cell types were consistently found nearer to these T cells, such as other lymphocytes, dendritic cells and Type 1 macrophages (Fig. 5c and Supplementary Fig. 6d, e). Other cell types were found further away at baseline, but the average distance was reduced when the samples were treated with MRTX1257, such as for tumour cells and Type 2 macrophages—in agreement with the relocation of T cells into the tumour domain. Looking in more detail at the cell types found in close proximity (within 100 pixels), some interesting patterns emerged (Fig. 5d). T cells were found closely located next to dendritic cells, in particular in the interface domain, an interaction that remained in the MRTX1257 treated samples. Regulatory T cells also showed a tight interaction with CD4+ T cells and seemed to be the main T cell type associating with cDC1 dendritic cells. In contrast to regulatory T cells, CD4+ and CD8+ T cells were found close to endothelium in the vehicle-treated images, but this association became less tight in the MRTX1257 treated condition. Type 1 macrophages were also found in proximity to the T cells in the

interface domain, but their interaction was more often indirect, at a peak distance of 20–60px from the T cells, which agrees with the earlier observation that they primarily interact with dendritic cells (Fig. 4d). Similarly, there was the presence of tumour cells in the neighbourhood of T cells in the interface, but mostly at a distance. The most striking change induced by treatment with MRTX1257 was the dominant interaction of T cells with Type 2 macrophages in the tumour domain, where also the tumour cells were found at a closer range. This result suggests that T cells will come under the stronger influence of tumour cells and Type 2 macrophages with treatment, potentially relating to their upregulation of PD-1.

**Tumour infiltration is accompanied by restoration of tumour vasculature.** To use a more unbiased approach to dissecting the differences between vehicle and MRTX1257 treated samples, we performed linear regression analysis on the mean intensities of all markers in the samples (Supplementary Table 4). Markers that were upregulated with MRTX1257 treatment included the macrophage maturation markers CD206, MHC-II and F4/80, as discussed above. Interestingly, vimentin was the fourth highest coefficient in the table. Also in the PCA analyses of Supplementary Fig. 4c, vimentin was one of the major variables contributing to components 1 and 2 (Fig. 6a), and vimentin was upregulated with MRTX1257 treatment across the ROIs (Fig. 6b). Vimentin expression is the canonical marker of epithelial-to-mesenchymal transition (EMT), which has been suggested as one of the possible mechanisms of resistance to KRAS inhibition[33,34]. In fact, the mean expression of vimentin increased for all cell types in the dataset, yet most striking was the difference seen for epithelium and endothelium (ratio: 4.7, $p$-adj = 0.02 and ratio 2.7, $p$-adj = 0.04, respectively, according to a linear mixed-effects model) (Fig. 6c). While vehicle-treated samples showed relatively low expression of vimentin on both cell types, following MRTX1257 treatment endothelium became one of the highest expressing cell types and this expression was predominantly within the interface and tumour domain (Supplementary Fig. 7). Upregulation of vimentin on endothelium can be a reflection of endothelium-to-mesenchymal transition[35]. However, a close inspection of the PECAM and vimentin expression in the tissues showed only modest colocalisation of the two antibodies at the pixel level. Vimentin is also a marker for pericytes, and while we do not have another marker in the panel that would

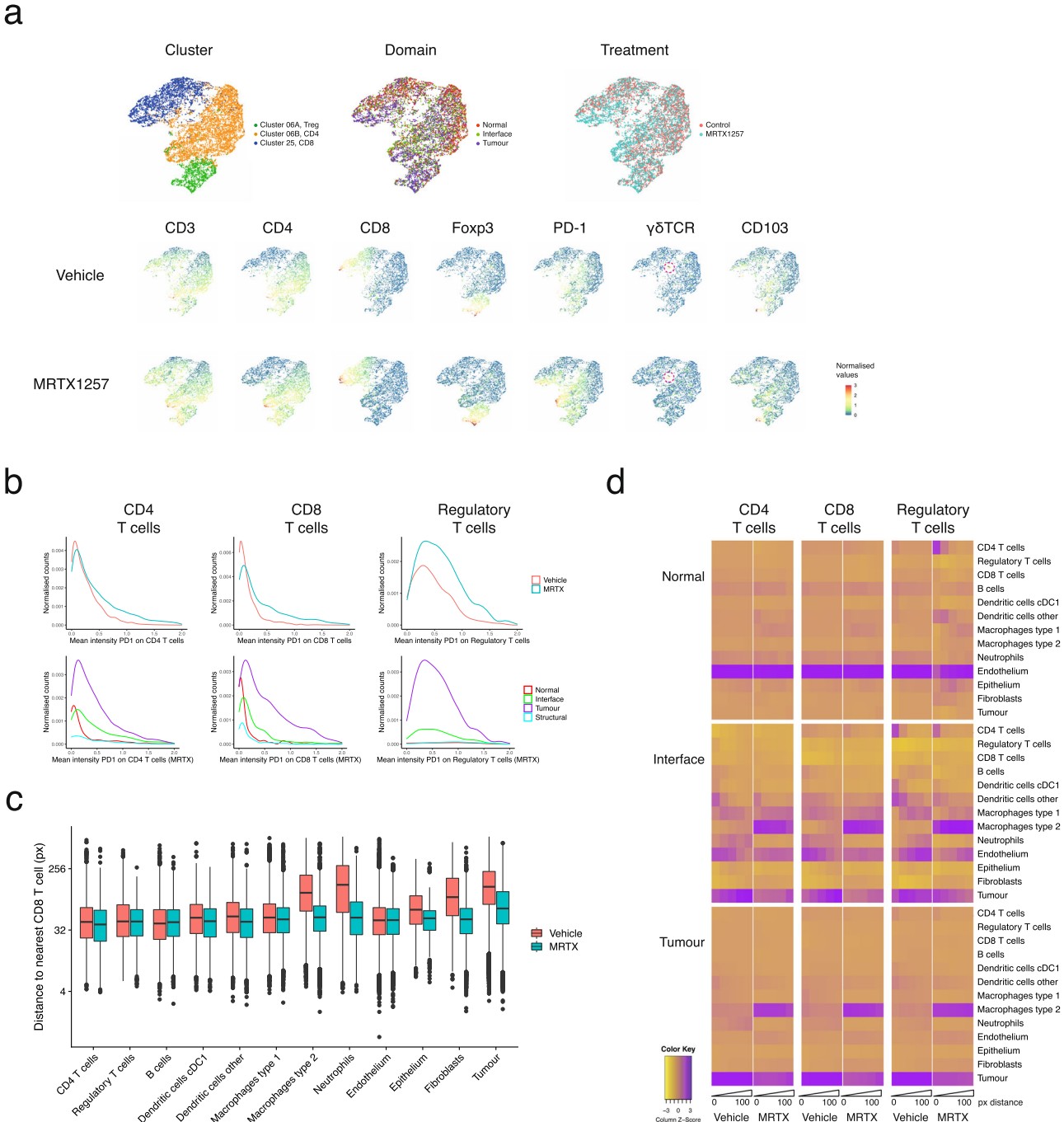

**Fig. 5 Phenotypic and spatial characterisation of T cells in response to treatment. a** uMAP of T cell clusters (top left), coloured by tissue domain (top middle), treatment (top right) and the expression of individual markers that were selected for the generation of these uMAPs in the bottom two rows, data separated by treatment group. **b** T cells upregulate PD-1 upon MRTX1257 treatment, but only those within the tumour domain. Histograms plotting the normalised counts for the mean intensity of PD-1 per CD4+ CD8+ or regulatory T cell. Top: data separated by treatment, bottom: MRTX1257 treated samples only, separated by tissue domain. **c** Box-and-whiskers plot depicting the distance of cell types to the nearest CD8+ T cell. In vehicle-treated tumours, CD8+ T cells are on average found in the proximity of other lymphocytes, endothelium, DCs and Type 1 macrophages, but not close to tumour, fibroblasts or Type 2 macrophages. Upon MRTX1257 treatment, these distances generally become shorter, reflective of migration of CD8+ T cells into the tumour domain accompanied by an increased presence of Type 2 macrophages and fibroblasts within the tumour. Boxes minima and maxima represent 25th and 75th percentile, centre depicts the median, whiskers indicate 1.5× interquartile range, dots are individual outliers. Linear mixed-effects modelling confirmed that between treatments the cell types display a significantly different distribution of distances to a nearest CD8 T cell ($P$ value = 0.0001 for treatment:cell type interaction, ANOVA tested). **d** Heatmaps for the different domains displaying cells in close proximity to the cell type of interest, separated by treatment and binned in distances from the centre of the cell (bins: 0–20px, 20–40px, 40–60px, 60–80px and 80–100px). Scaled on columns to show the relative contribution of cell types to the neighbourhood. MRTX MRTX1257, px pixels.

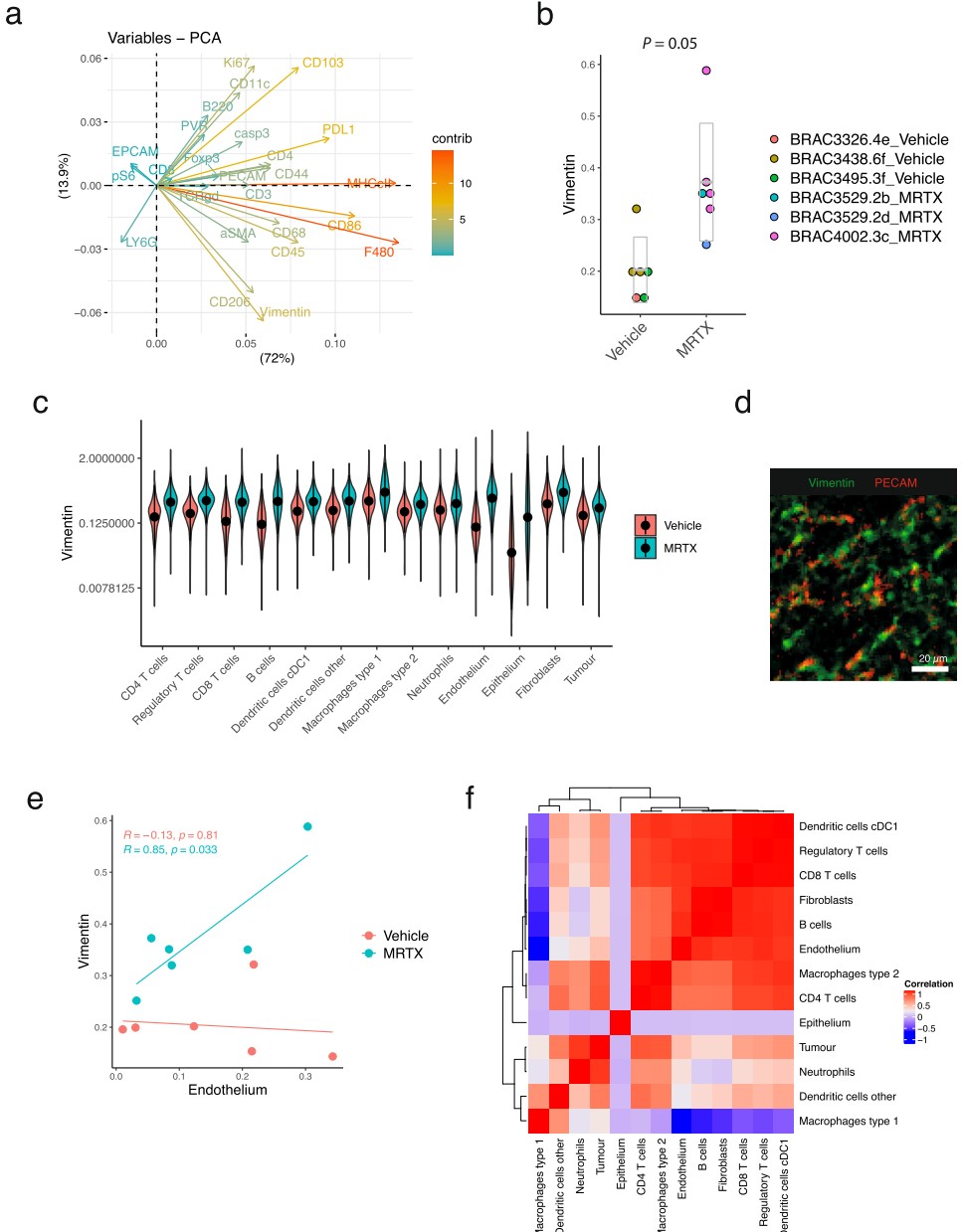

**Fig. 6 MRTX1257 treatment is correlated with increased vimentin expression and tumour vascularisation. a** Plot depicting the contribution of the mean intensity of markers per mouse that have contributed to the first two principal components (PC1 and PC2 on and *x* and *y* axis, respectively) of the PCA analysis from Supplementary Fig. 4c. **b** Mean intensity of vimentin per ROI, separated by treatment group on the *x*-axis. Points are coloured by mouse ID as indicated in the legend. *P* value = 0.03, extracted from ANOVA testing of linear mixed-effects model. **c** Distribution of vimentin expression across cell types separated by treatment, means are indicated with a dot. ANOVA testing of a linear mixed-effects model found that the two treatment conditions had a significantly different vimentin expression profile across the cell types (*P* < 0.0001 for treatment:cell type), and that these differences were significant for Endothelium (*P*-adj = 0.04) and Epithelium (*P*-adj = 0.02, corrected for multiple testing by adjusting for false discovery rate (FDR)). **d** Crop of PECAM and vimentin expression in a tumour treated with MRTX1257. For visualisation purposes, the images were processed in Fiji with a median filter (radius 0.5). **e** Vimentin expression and proportion of endothelium cells in the tissue are positively correlated for the MRTX1257 treated ROIs, but not for the vehicle-treated ROIs (two-sided Pearson correlation testing, not adjusted for multiple testing). No other correlations between cell type and vimentin expression were found to be significant. **f** Pearson correlation matrix of cell type proportions in the tumour domain of MRTX1257 treated tumours, extracted from the linear mixed-effects model using the mouse:ROI estimates. MRTX MRTX1257.

unambiguously identify these cells, the observed expression patterns strongly suggest the presence of pericytes lining the endothelium (Fig. 6d). Indeed the expression of vimentin correlated with the proportion of endothelium in the MRTX1257 treated samples (Fig. 6e). Together with the increased presence of endothelium in the tumour (Fig. 3c, d), the detection of pericytes points to normalisation of the vasculature of the tumour as a

result of KRAS G12C inhibition. Improved vascularisation of the tumour agrees well with the observation that the MRTX1257 treated samples harboured fewer necrotic patches in the tumours, but can also have played a role in facilitating tumour infiltration by effector cells, as the proportion of endothelium in the tumour correlated with the level of infiltration by T cells and dendritic cells in the MRTX1257 treated tumours (Fig. 6f).

## Discussion

Multiplex imaging techniques such as IMC are beginning to take centre stage in studies of the TME[36]. The application of this technology to preclinical mouse studies, however, has been lacking. Here we have described the design, optimisation and validation of a complete IMC workflow for mouse tissues. Our panel of 27 antibodies provides a good basis for immuno-oncology studies, identifying many of the cell types of interest in the TME. Additional isotopes are still available to allow for customisation in any area of research interest, such as additional checkpoint molecules or stromal markers, or for combination with RNAscope as was previously shown for human FFPE tissues[37].

Considering that frozen tissues generally have a reduced quality of histomorphology compared to FFPE and the 3LL ΔNRAS cell line gives rise to very densely packed tumours, we set out to find an optimised segmentation method, critical for obtaining good quality single cell data. We tested modifications to previously published image segmentation methods and established two strategies with a better performance with respect to signal-to-noise ratios and cell identification. The one-pixel expansion segmentation can represent a simple strategy to obtain quick segmentation data from a new antibody panel, without the need for training the classifiers. More flexible and sensitive segmentation can be obtained with a sequential segmentation strategy and importantly, this method can provide a solution to cells that are more challenging to segment. Any of those segmentation methods can simply be applied to large datasets using our automated image segmentation pipeline, which has been made available to the wider community via the nf-core platform (nf-core/imcyto). This pipeline is scalable, yet flexible, as it can be customised to work with any antibody panel or segmentation strategy.

Working with mouse tissues has the advantage of imaging a cross-section of the whole tumour, which is generally not possible with human samples due to the significantly larger size. This provides enhanced insight into the tumour architecture, as we have seen here for the very distinct cell communities within the different areas of the tissue, such as the accumulation of effector cells and the striking spatial separation of two distinct phenotypes of macrophages. We found Type 1 macrophages expressing CD68 and CD11c, resembling normal or tumour-associated tissue-resident alveolar macrophages as previously described[27,29,38], though lacking expression of mannose receptor CD206. These cells accumulated at the tumour edge whilst they were being excluded from the main body of the tumour. In contrast, Type 2 macrophages were intermixed with the tumour cells and were characterised by expression of F4/80 and CD206. These macrophages possibly represent tissue-resident interstitial macrophages or may have been recently recruited from the circulating monocyte pool, as previous studies showed that circulating classical monocytes could differentiate into F4/80-high tumour-associated macrophages in a model of metastatic breast cancer in the lung[39]. Casanova-Acebes and colleagues recently characterised the lung macrophage populations on both human and mouse NSCLC using single cell transcriptomics, similarly separating the macrophages in two large clusters, one (I) with an alveolar macrophage profile like our Type 1 macrophages and also located at the tumour periphery, and a second monocyte-derived population (II) more similar to our Type 2 tumour infiltrating macrophages[38]. Chakarov, et al. described the existence of two interstitial macrophages subsets, Lyve1[hi] CD206[+] and Lyve1[lo] MHC-II[+], both derived from monocytes[40]. While the Type 2 macrophages in our data aggregate in the uMAP as one expanding population with increased expression of activation markers upon MRTX1257 treatment, there is some differential

expression of CD206 and MHC-II within this subset that could be consistent with the recruitment of two types of interstitial macrophages. Zilionis, et al. used single cell transcriptomics to distinguish four macrophage and three monocyte subsets in a mouse lung cancer model[3], a granularity that our IMC panel cannot offer[3]. Technologies that couple transcriptome analysis with cell surface marker expression such as CITE-seq will aid us to better match the macrophage phenotypes observed here to their transcriptional activation state.

While MRTX1257 is a tumour-specific inhibitor, acting only on the G12C mutant form of KRAS protein found in the tumour cells and not the wild-type KRAS protein found in the cells of normal tissue, it had profound indirect effects on the TME. The studies of Canon et al[15]. and Briere et al[16]. used flow cytometry and immunohistochemistry to look at changes in the TME upon KRAS G12C inhibition in the immunogenic subcutaneous colon cancer model CT26. The main changes they observed were increases in macrophages, DCs and T cells. Our data paralleled those observations in an immune evasive orthotopic lung cancer model and provided additional spatial phenotypic data. We have used several different approaches to explore the spatial relationships between cell types, within cell neighbourhoods, and across the tissue domains. We showed how the tumour bed becomes infiltrated with maturing macrophages and PD-1[+] lymphocytes, changing the spatial interactions between these cells. Interestingly some cells seemed to consistently exist together, such as the Type 1 macrophages that remain within the cell neighbourhood of DCs, irrespective of whether they are in the interface or in the tumour domain. This suggests either that their recruitment is regulated by the same mechanism, or that there is functional importance to their interaction. While the DCs in that cellular niche expressed co-stimulatory molecules such as CD86 and could therefore potentially activate the T cells, the Type 1 macrophages expressed high levels of PD-L1 and would, in turn, be able to repress the T cell activation. On the other hand, CD4 T cells in the interface of vehicle-treated samples interacted strongly with regulatory T cells that can play a role in suppressing T cell activation; this interaction was lost in the MRTX1257 treated samples as the number of regulatory T cells in the tumour did not increase in all but one of the tumours assessed. Clearly, such local interactions could be fundamental to the outcome of immunotherapeutic treatments. It also raises the question of the state of the local chemokine and cytokine milieu. While a study by Schulz, et al. demonstrated that IMC can be combined with transcript detection using an RNAscope based in situ hybridisation[37], the detection level of cytokine mRNA is often a limiting factor. A multi-omics approach combining IMC with single cell approaches such as CITE-Seq[41], or spatial transcriptomics[42,43] would therefore have the potential to significantly enhance our insight into the mechanisms regulating cellular communities.

A major advantage of multiplex technology is that it allows for discovery and hypothesis generation. The finding of a higher expression of vimentin across all cell types was unanticipated but could be explained by greater mobility of cells in the remodelling process of the TME[44]. In addition, it led to the observation that there was an increased presence of endothelium and possibly pericytes, pointing towards normalisation of the tumour vasculature upon MRTX1257 treatment in this normally haemorrhagic and necrotic tumour model[45]. Altogether, our data shows that treatment with a tumour-specific KRAS inhibitor can dramatically remodel the TME in favour of antitumour immune response, even in an immune cold tumour model such as the 3LL Lewis lung carcinoma. Current investigations are aimed at elucidating the mechanism by which this conversion is mediated.

We have presented here a complete IMC workflow for use in preclinical mouse studies and demonstrated the value and importance of using IMC to study the effects of treatment on the spatial organisation of the TME.

## Methods

**In vivo drug study**. This work was performed under a UK Home office approved project license and in accordance with The Francis Crick Institute welfare guidelines. Mice were bred and maintained in specific-pathogen-free (SPF) conditions, housed up to 5 per cage in individually ventillated cages (IVC), with a 12–12 h light–dark cycle. Food and water were provided ad libitum. $10^6$ 3LL ΔNRAS Lewis lung carcinoma cells[17] were injected in the tail vein of 9–11-week-old C57BL/6 mice and allowed to establish for 3 weeks. The lung tumour burden was assessed using a Quantum GX2 microCT Imaging System (PerkinElmer) and mice were assigned to treatment groups of similar tumour burden. MRTX1257 was prepared by sonication in 10% Captisol® (Ligand) and 50 mM citrate buffer (pH 5.0) and administered daily at 50 mg/kg by oral gavage (5 μl/g) for 7 days. Four hours after the last treatment mice were scanned again and sacrificed with a terminal overdose (0.1 ml/10 g body weight intraperitoneal) of a mixture of Pentobarbital (2% w/v) and Mepivacaine hydrochloride (8 mg/ml), followed by cervical dislocation.

**Tumour volume measurements**. Mice were scanned 1 day before start of treatment and on the last day of treatment. Mice were anesthetised by inhalation of isoflurane (Abbott Labs) and CT scanned using the Quantum GX2 microCT imaging system (Perkin Elmer). Breathing rate and body temperature were measured throughout the scan using in-built physiological monitoring devices. Scanning parameters were as follows: copper and aluminium filter 0.06 mm + 0.5 mm, respectively, 1° rotation step over 360°, source current 40 μA, source voltage 90 kV, image isotropic pixel size 50 μm. Scan mode at High Speed & Gating 4 min. Gating technique set at respiratory gating. Lung images were grouped into bins based on the respiratory cycle and images reconstructed using the Quantum GX2 programme with parameters set at Acquisition FOV 36 mm and Recon FOV 25 mm. Estimations of lung tumour volumes were generated by highlighting 3D ROIs in the imaging programme Analyze, version 12.0, from AnalyzeDirect.

**Tissue processing**. Three mice from both the vehicle and the MRTX1257 treated group were processed to be used in IMC. Dissected lungs with tumours were stored in 20% ice-cold sucrose up to 1 h before embedding in Tissue-Tek O.C.T. Compound (Sakura) and being frozen gently using an isopentane liquid nitrogen bath. Blocks were stored at −80 °C until further processing.

**Antibody staining**. Five-micrometer thin tissue sections were cut and collected onto SuperFrost Plus™ Adhesion slides (Thermo Scientific) in a cryostat and stored at −80 °C. When required, slides were thawed for 3 hours, fixed for 10 min with Image-iT™ Fixative Solution (ThermoFisher) and washed with DPBS (GIBCO) and then DPBS/0.05%Tween. Slides were blocked with Superblock blocking buffer in DPBS (Thermo Scientific) for 30 min and then 1:100 anti-CD16/CD32 Fc-block (BD Biosciences) for 10 min. Staining was done with a cocktail of primary antibodies in 0.5% BSA and 1:100 anti-CD16/CD32 Fc-block in DPBS/0.05%Tween for 1 h in a dark humid chamber and followed by washes in DPBS/0.05%Tween and then DPBS alone three times each and rinsed in MilliQ water. For IF the samples were stained with three dilutions (1:40, 1:100, 1:200) of the primary antibody, washed as above but not in water and then mounted with ProLong Gold Antifade Mountant with/without DAPI (Thermo Scientific), cover-slipped and left to air dry overnight. For IMC, after the final DPBS wash slides were incubated with 1:500 Cell-ID Intercalator-Ir (Fluidigm) in DPBS for 10 min and rinsed with MilliQ water before air-drying overnight. Antibody clones conjugated with metal isotopes were purchased from Fluidigm where available (mouse CyTOF catalogue) or obtained in purified format and conjugated in house using MaxPar conjugation kits (Fluidigm) according to the manufacturer's protocol. Details of the antibodies and dilutions used for IF and IMC are listed in Supplementary Tables 1 and 2.

**Image acquisition**. IF slides were imaged with a Zeiss Upright LSM710 microscope with a 20x objective lens using Zen blue imaging software (Zeiss). IMC images were acquired using Hyperion Imaging Mass Cytometer. Each ROI was selected such that it would contain a whole tumour including adjacent normal tissue where possible, or, if required, they were cropped post-acquisition to contain a single tumour. Twelve images obtained from six mice were selected for this study, ranging from 1–9 mm² (429 Mb up to 3.35 Gb).

For figures in this publication with IF or IMC images we used Fiji ImageJ v2.0.0 to make composite images of selected channels. For visualisation purposes the images were processed with an outlier removal step, filtered using a median or Gaussian filter (0.5px radius) and scaled to enhance contrast. For Fig. 1c the lymphocyte images were filtered using a band pass (3–40px) Fast Fourier Transformation to enhance detection of the cells.

**Segmentation pipeline**. The following section describes the proposed segmentation pipelines carried out in CellProfiler v3.1.9, including custom modules by Bodenmiller (https://github.com/BodenmillerGroup/ImcPluginsCP) and Ilastik v1.3.3b1 (see Code availability and Supplementary Fig. 3). In brief, data obtained as.txt files were converted into stacks of individual.tiff files per antibody marker using the IMCtools package v1.0.5 (https://github.com/BodenmillerGroup/imctools)[20]. Images in the full stack path were minimally filtered using outlier removal and median filtering in CellProfiler (both using the custom "Smooth Multichannel" module). For nuclear segmentation, images for 191Ir and 193Ir were summed, histogram equalised and segmented using a propagation-based thresholding strategy in CellProfiler. For the pixel expansion strategy, segmented nuclei were expanded by a set number of pixels (1 or 3) to identify whole-cell objects (see Fig. 2a).

For the propagation strategy, selected markers (CD44, CD45, PECAM, CD11c, CD3, CD4, CD68, CD8, F480, aSMA, B220 and CD206) were merged, alongside a nuclei image and converted into an RBG composite tiff in CellProfiler. This composite was fed into Ilastik pixel classification workflow and classified into nuclei, membrane and background to generate probability maps. In CellProfiler both segmented nuclei (as described) and membrane probability maps were used for whole-cell segmentation using a propagation-based thresholding strategy (see Fig. 2a).

Similarly, for the sequential segmentation strategy, multiple images representing individual cell types and domains were created by merging together selected markers in CellProfiler (PECAM for normal, CD44 for tumour, EPCAM and aSMA for structural, CD3, CD4, CD8 and B220 for lymphocytes, and CD68, F480, CD206 and CD11c for macrophages). These five new images alongside a nuclei image were combined into a six-channel image in CellProfiler. This was fed into Ilastik autocontext workflow and classified into tissue domains (nuclei, tumour, normal, structural) and cell types (lymphocytes, macrophages, fibroblasts) to generate probability maps. In CellProfiler, individual cell types (in order—lymphocytes, macrophages, fibroblasts, normal cells, tumour cells, remaining cells) were sequentially segmented using both segmented nuclei (as described but with customised size parameters) and corresponding probability maps, with the exception of fibroblasts which were identified only using the probability map. All cell types were added together to generate a total cell mask. Domain segmentation was performed using thresholding on domain probability maps, creating normal, tumour and structural domains. The interface domain was created from the overlap between normal and tumour domains. For CellProfiler and Ilastik project files used in these segmentation strategies, see Data availability.

**Validation of segmentation strategies**. All cells in the ROI "BRAC3438.6-f_ROI1_t1_Vehicle" (in short: "02_Vehicle"), were ranked for expression of each of the markers depicted, and the top 500 highest expressing cells were selected to calculate the mean intensity for that marker within the top 500, and the remaining cells, as a measure of signal enrichment. Signal-to-noise ratios were calculated by taking the mean intensity of relevant marker in the top 500 cells, relative to the expression of "polluting" markers not expected to be expressed in these same cells. We generated a small manually annotated dataset from the same image set as above, by recording the $x$ and $y$ coordinates of CD4+ ($n = 92$) and CD8+ T cells ($n = 70$), CD103+ DCs ($n = 65$), F4/80+ macrophages ($n = 108$) and aSMA+ fibroblasts ($n = 88$). Cells from the top 500 expressing CD4, CD8, CD103, F4/80 or aSMA were interrogated to determine how many cells of each matched up to cells in the annotated dataset, following the criterium that the cell centre had to fall within 5px of each other.

**imcyto pipeline**. An automated Nextflow-based pipeline that sequentially pre-processes and segments imaging data and extracts single cell expression information (see Supplementary Fig. 2). This pipeline combines Docker or Singularity containers for IMCtools, CellProfiler and Ilastik. Inputs for the pipeline are image data in the form of .mcd, .txt, or ome.tiff files, a .csv file with binary information on the channels to be included, the user-generated CellProfiler pipeline files (.cppipe) - custom CellProfiler modules need to be separately provided, and (pretrained) Ilastik project files (.ilp). Output files from each step in the pipeline, such as raw or preprocessed TIFFs, probability maps, masks, Object relationships and measurements in the form of a .csv file will be created in an organised results folder structure. Alongside, a report is generated on the performance of the pipeline, with respect to memory and processor usage and running time for each individual processing step.

A detailed user guide can be found on: https://github.com/nf-core/imcyto

**Data normalisation, scaling and clustering**. Data for each ROI was normalised to mean intensity of Xenon134. To create the larger dataset, data from six control tumour ROIs and 6 MRTX1257 treated tumours were concatenated (282,837 cells in total for all 12 images combined) and channels scaled to the 99th percentile. A few small image areas containing a sudden drop in counts were excluded from the analysis, by excluding cells with X and $y$ coordinates falling within those areas. Mean pixel intensity of 17 specific cellular markers from the data were selected for high-dimensional, unsupervised clustering using Rphenograph:[23] αSMA, B220, CD103, CD11c, CD3, CD44, CD45, CD4, CD68, CD8, EPCAM, F480, LY6G,

MHC-II, NKp46, PECAM and PVR. This grouped the single cell data into separate subpopulations based on phenotypic similarity of the chosen markers and assigned each to a cluster number using the Louvain community detection algorithm. A k input of 20 generated 30 clusters, and further manual separation into subclusters by expert gating resulted in 35 distinct subpopulations overall. Clusters were evaluated to associate each with a cell type based on the distribution of weighted expression of each marker and grouped into 13 cell types based on their phenotypic classification (see Supplementary Table 3).

**Phenotypic analysis**. R implementations of tSNE (RtSNE)[46] and uMAP[26] were used for dimensionality reduction. Macrophage uMAP analysis was based on clusters 05A, 08, 11, 21A, 23 and 26, using the markers MHC-II, F4/80, PD-L1, CD86, CD68, CD206, pS6 and CD11c. The T cell uMAP was based on clusters 06A, 06B and 25, using the markers CD3, CD4, CD8, TCRgd, Foxp3, PD-1 and CD103.

**Spatial analysis**. A permutation approach developed by the Bodenmiller lab was used to determine whether detected neighbour interactions between cell types occurred more frequently in the images than observed by randomisation (https://github.com/BodenmillerGroup/neighbouRhood)[19]. Briefly, an 'objectTable' logging object, and cluster information, and 'Object relationships' listing each cell and all its identified neighbours within a 15-pixel distance, were sent through 5000 rounds of permutation. A P value was then calculated to determine whether the difference between average baseline and permuted neighbour interactions were significantly different for each pairing combination of cell types. Significant interaction or avoidance was determined by baseline values falling above or below the distribution of permutated data, respectively. Averaged interactions for baseline versus permutation statistics from the Bodenmiller neighbouRhood analysis script were then compared for each pairing of cell types from every image by calculating log2 fold chance (log2FC) to determine how treatment affects neighbour interactions. For Type 1 and Type 2 macrophages, the log2FC was also calculated for normal, tumour and interface domains separately.

The Pythagoras's theorem was used to compute distances of each cell in the dataset to the nearest CD8$^+$, CD4$^+$ and regulatory T cell, based on the $x$ and $y$ coordinates of the centre of the cells.

**Statistics**. We used the 'lme4' package within R to fit a mixed-effects model to account for fixed effects of domain, cell type and treatment, whilst allowing for a per-mouse and ROI-within-mouse variation of the distribution of cells between cell types. We use a Poisson model as a surrogate to fit the multinomial distribution of cell counts across the cell types. Individual comparisons were carried out using a Wald test. We used the correlation structure between ROIs in the same treatment group, as predicted by mixed-effects models fitted on the treatments separately to build the correlation heatmaps of cell type proportions. Similarly, the mean intensity of vimentin was fit with a mixed-effects model to describe the variation of vimentin expression between cell types and treatments. Other statistics as specified in the figure legends.

**Reporting summary**. Further information on research design is available in the Nature Research Reporting Summary linked to this article.

## Data availability

The raw image files, pipeline input files, processed single cell data, 3D plot and associated files generated and used for this study, are available in a Figshare repository at https://hdl.handle.net/10779/crick.c.5270621.

The remaining data is available within the Article and Supplementary Information.

## Code availability

The automated imcyto image segmentation pipeline built using Nextflow is freely available at https://nf-co.re/imcyto and https://github.com/nf-core/imcyto[47].

The code generated during this study for analysis of single cell IMC data following image segmentation was written for R v3.6.2 and is available at Figshare: https://hdl.handle.net/10779/crick.c.5270621 and https://github.com/FrancisCrickInstitute/vanMaldegem_Valand_2021[48].

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

## Acknowledgements

We thank David Barry, Leigh-Anne McDuffus, Richard Mitter and Julia Handl for helpful discussions. We thank the science technology platforms at the Francis Crick Institute including Biological Research Facility, Scientific Computing, Bioinformatics and Biostatistics, Flow Cytometry, Experimental Histopathology, Advanced Light Microscopy and Cell Services. This work was supported by funding to J.D. from the Francis Crick Institute—which receives its core funding from Cancer Research UK (FC001070), the UK Medical Research Council (FC001070), and the Wellcome Trust (FC001070)—from the European Research Council Advanced Grant RASImmune, from a Wellcome Trust Senior Investigator Award 103799/Z/14/Z and from a Cancer Research UK Cancer ImmunoTherapy Accelerator Award (CITA-CRUK; C33499/A20265). K.E. is supported by the European Union's Horizon 2020 research and innovation programme under the Marie Sklodowska-Curie grant agreement No 838540 and research grant funding from Bristol Myers Squibb.

## Author contributions

F.v.M., K.V., M.C., M.M.-A., C.S. and J.D. designed the study, interpreted the results and wrote the manuscript. F.v.M., K.V., M.M.-A., V.S.K.T. and S.R. performed the biochemical experiments. G.K. contributed to the statistical analysis. C.M. and E.M. assisted with in vivo studies. F.v.M., K.V., M.C., H.P., N.B., M.A. and E.C. performed computational analyses and set up the pipeline. P.H. and D.L. carried out IMC analysis. M.A., K.E. and E.C. provided image analysis expertise. All authors contributed to manuscript revision and review.

## Competing interests

J.D. has acted as a consultant for AstraZeneca, Bayer, Jubilant, Theras, Vividion and Novartis, and has funded research agreements with BMS, Revolution Medicines and Boehringer Ingelheim. C.S. receives grant support from Archer Dx, AstraZeneca, Boehringer Ingelheim and Ono Pharmaceutical; has consulted for AstraZeneca, Bicycle Therapeutics, Celgene, Genentech, GRAIL, GSK, Illumina, Medicxi, MSD, Novartis and the Sarah Cannon Research Institute; receives grant support and has consulted for Bristol Myers Squibb, Pfizer and Roche–Ventana; is an advisory board member and is involved in trials sponsored by AstraZeneca; has stock options in Apogen Biotechnologies, Epic Sciences, GRAIL; and has stock options and is a co-founder of Achilles Therapeutics. The remaining authors declare no competing interests.
