## [Peer Review File · Nature Communications]

REVIEWER COMMENTS

Reviewer #1 (Remarks to the Author): Expert in multi-plex imaging

Van Maldegem and colleagues report on an imaging mass cytometry panel and analytical tools for mouse tissue and, in parallel, have studied spatial immunophenotypical alterations related to the use of MRTX1257 on a KRAS-mutated mouse tumor model. The reporting of a ready-to-use IMC panel for mouse tissue together with an interesting cell segmentation approach are the main contributions of the paper. The most relevant immunological changes associated to the treatment with MRTX1257 have already been reported elsewhere.

Main comments:

1. I think the presentation of the results should be simplified. There is currently a mixture between methodological and biological questions throughout the paper that do not facilitate its reading. I think the authors should, first, solely focus on the methodological developments implemented by them and then move to their application in the chosen model. In line with this, I think the Results section can be reduced considerably.
2. The choices made to visualize the data in Figure 3 are not the most useful ones. For instance, in Figure 3a, it is very difficult to evaluate the contribution of each marker for the definition of a particular subset. Re-scaling marker expression per subset might be helpful?
3. I think the data has been over-clustered when considering the amount of markers that have been employed and when looking at Figure 3b. Some clusters should be merged in case they do not represent biologically
4. The authors mention PD-1 as being expressed mainly by CD8+ T cells, but in Supplementary figure 5e, PD-1 expression seems to be mainly derived from CD4 T cells?

Minor comments:

1. Is CD44 really a bonafide stem-cell marker when considering that all tumor cells in Figure 1C are staining for it? Isn't CD103 also a marker for intraepithelial/tumor-resident T cells? MHC-II, general activation marker?
2. Is CT26 IMC data available? It would be interesting to see whether, despite the lack of response to the inhibitor, immune cell infiltration was nevertheless promoted.
3. ""Treatment of 3LL NRAS tumours established (...) although most tumours did not actually reduce in size (Supplementary Figure 2)."" The lack of response in CT26 should also be included in the figure as this sentence mentions a comparison with control mice.
4. ""Vehicle treated tumors usually showed a tissue architecture..."" I think on of the most striking features in addition to the accumulation of macrophages in the surrounding of the tumor is the infiltration by Ly6G-positive cells. This should have been highlighted?

5. "Macrophage phenotypes changed" – this expression is too ambiguous as the authors probably cannot distinguish a change in phenotype from a de novo infiltration by other subsets.

6. Figure 1C – it's very difficult to distinguish CD11c and CD68, I suggest that CD11c is removed or that another panel is included.

Reviewer #2 (Remarks to the Author): Expert in segmentation methods

The authors present an imaging mass cytometry (IMC) workflow to analyze mouse tumors, suitable for preclinical research. The workflow contains a good preselection of 27 antibodies targeting typical markers and can be extended using 12 custom antibodies. The authors image lung tumors from 3 mice of vehicle and 3 mice of treatment group and analyze the data. They compare 4 spatial cell segmentation algorithms and evaluate their properties. Cells are classified into types using clustering with manually adjusted gates. Based on the cell classification, the tissue is segmented into domains (e.g. tumor vs. healthy tissue) and cell populations are assessed. Furthermore, cell-cell distances, e.g. between tumor and CD8+ cells are quantified and some example marker expression profiles are analyzed. Overall, the imaging technology and analysis software is convincingly showcased using a relevant mouse model. The software (and data) is published with the paper and the solution based in Nextflow appears to be highly reproducible. The provided tool seems to be of potentially high value for preclinical research and is claimed to be the first of its kind. I think that this work is a valuable contribution but several issues need to be addressed:

Introduction

At the end of the introduction, I am missing a statement about the aim of the study, giving a rationale and overview for the following sections.

Results

Section "An antibody panel for imaging mass cytometry on mouse tissues"

The functionality was nicely shown in a spleen follicle.

Why are there 12 free slots if 27 are taken and 40 are available?

Section "Targeted inhibition of KRAS G12C in a preclinical model of lung cancer"

While the paper is well-written in general, with good figures, I had some trouble understanding the experimental design. It should be made clear at the beginning that there are two groups (vehicle-treated and MRTX1257-treated), with 3 mice each. The sentence "A total of twelve tumours were

selected for IMC, six tumours from three mice of each treatment group" is too unclear and I had to read the whole paper several times to figure it out.

Is "Orthotopic transplants" the right term for tumor induction by tumor cell injection?

The mention of the CT26 tumor is confusing here, as it conveys the impression that you had an experimental CT26 group you are referring to.

The discussion of F4/80+ and CD206+ macrophages and CD4+ and CD8+ T cells is a bit confusing here, because there is a much more thorough discussion later.

Section "Optimisation of single cell segmentation using cell type specific settings"

Since this seems to be an important innovation or part of the tool, this should be mentioned in the outlook at the end of the introduction.

Please briefly mention the scope or purpose of the libraries CellProfiler and Ilastik.

Section "IMC single cell analysis reveals dramatic remodelling of TME"

This is a interesting section but it is too long. Please break it down, maybe into clustering, domain analysis, cell-cell-interactions, macrophages, lymphocytes and marker-expressions (vimentin).

What I lack most in the results (and figures) is a statistical analysis comparing the two groups, e.g. that the mean distance to of tumor to CD8+ cells decreased significantly in the treatment group compared to the untreated group. This means that for each mouse one score is generated, we have three scores per group (3 mice per group) and a test (e.g. t-test) showing a significant difference for the 3 vs 3 values. Each mouse is a sample, not each tumor. Such (simple) statistical comparisons between groups are widespread in preclinical research and therefore would be interesting for the intended audience which is supposed to apply the proposed tool. Such an analysis would also give an impression of the inter-group variance of such scores.

Discussion:

Please mention some the differences between human and murine cell images. Are mouse cells/nuclei smaller?

When saying "Seeing the tumour as a whole..." please mention that you still analyze a single slice only.

Methods:

Please provide some details about the μ CT scan, e.g. voltage, power, exposure, projection size and count, voxel size.

Please explain: "treatment groups were randomised with stratification by tumour number and size"

Figures:

Figure 2: a) please mention sequential segmentation. Maybe put panel b at the beginning.

Reviewer #3 (Remarks to the Author): Expert in tumour microenvironment

Van Maldegem, Valand et al. developed a useful automated pipeline for the analysis of imaging mass cytometry (IMC) data obtained from frozen mouse tissue sections. Availability of such pipelines is highly valuable for the unbiased structural evaluation of the tumor microenvironment (TME). The authors refined existing tools previously developed by Bodenmiller and others (largely developed for FFPE sections of human tumors) and improved the analysis of the spatial distribution of cells in the TME of frozen mouse tumors. However, it appears that the novelty of the approach is only incremental, as clearly acknowledged by the authors. In fact, the pipeline is largely based on algorithms developed by Bodenmiller and others (ImcPluginsCP and IMCtools for segmentation; Rphenograph and Louvain community detection algorithm; tSNE, uMAP for clustering; neighbourhood for spatial analysis and Slingshot pseudotime trajectory analysis; etc.). This said, the availability of the new workflow amenable to the analysis of frozen mouse tissues is certainly valuable and would be of broad interest.

Regarding the biological findings of the manuscript – the characterization of mouse lung tumors treated with a KRAS-G12C inhibitor – the results are descriptive and do not seem to add much to what could be obtained using alternative methods. Indeed, the main results are consistent with previous work (Canon et al. Nature 2019; Hallin et al. Cancer Discovery 2020; Adachi et al Clin Cancer Res 2020; Singh et al Cancer Cell 2009).

In summary, this is a useful technical report that will advance the application of IMC to frozen mouse tissues. However, at this stage of development, application of the refined method did not unravel new biology of KRAS-G12C inhibition.

Specific comments

1. Figure 3a: could authors explain the reasons for excluding some markers in the clustering? In fact, adding Foxp3 would have been important to identify regulatory T cells, which have very distinct biology compared to other CD4 T cells (re bulk of CD4 06 cluster).
2. Figure 3a and 4a: authors claim that cluster 11 corresponds to type 1 macrophages. However, it is not clear how they distinguished them from dendritic cells, as both display low F4/80 expression. It would be good to list the exact markers that were used for the definition of each cluster in a supplementary item (the current description is not exhaustive enough).
3. Figure 4d. Authors perform a Slingshot pseudotime trajectory analysis to assess the differentiation of macrophages. Such algorithms were created to delineate possible differentiation trajectories from high dimensional datasets like scRNAseq. However, in the case of this study only a few parameters were considered for such analysis. In fact, only 6 markers are relevant for macrophages (authors should clarify if only specific macrophage markers were used for the analysis or all markers were included). Therefore, the suitability of such algorithms and their advantages over standard characterization of different macrophage populations and expression of individual markers (CD68, MHCII, CD206, CD11c, PDL1, CD86) are apparently questionable. In addition, as expression of each marker was based on a segmentation that is not perfect, differences in expression could derive from technical errors that are not compensated by analysis of hundreds of variables (as it happens when analysing scRNAseq data). What are the results of performing the same Slingshot pseudotime trajectory analysis by separating treated and untreated tumors?
4. Figure 4e: Why are there different numbers of data points (presumably samples) for type1 and type2 macrophages in the neighbourhood analysis? Using the same scale for the analysis of type 1 and type 2 macrophages would make the graph easier to interpret.
5. Figure 4e: It may be interesting to assess the neighbourhood of CD8 T, CD4 T, T regs and B cells separately as it is known that macrophages can play distinct functions on different lymphocyte subtypes. Also, it would be interesting to perform such neighbourhood analysis on the tumor edge vs tumor core to reinforce the statement that "CD68+ macrophages are found often in close proximity to T cells and CD103+ DCs, particularly at the edge of the tumour (Supplemental Figure 5b). This is also true of the minority of intra-tumoural CD68+ macrophages, which maintain neighbourhoods with T cells and dendritic cells". Is it possible to assess whether the neighbourhood enrichment is significantly different in type1 vs type 2 macrophages? According to the graphs, it seems that log2FC enrichment values of type 1 and type 2 macrophages vs DCs are similar.

6. Figure 1B. CD4 is also a marker of regulatory T cells in addition to Th cells. CD44 is also a differentiation marker for T cells.

7. Figure 3a-c. What markers were used to identify normal lung (non-tumor associated) cell clusters? It would be good to list the exact markers that were used for the definition of each cluster.

Minor points

8. Supplementary Figure 4a-b. What do the grey bars represent? If the bar colouring is correct, it looks like there are no fibroblasts or macrophages in vehicle-treated mice and no neutrophils in MRTX-treated mice.

9. Supplementary Figure 4d: Is there an error in the image of 03_vehicle? There is no segmentation observed but a big purple ball.

10. Supplementary Figure 5d: It would be interesting to plot CD4 and CD8 T cells with different colours to see if they show distinct distributions.

11. In the introduction the authors state “it has become apparent that tumours are infiltrated with a diverse spectrum of immune cells, often with different phenotypes from their normally homeostatic counterparts 1, 2”. Ref 1 is not adequate as does not assess tumor-infiltrating cells.

12. Figure legends should define the sample size if it varies among panels.

12. In the sentence that states “greater expression of the maturation and exhaustion marker PD-1”, it would be more appropriate to say “activation and exhaustion marker PD1”, as naïve T cells are generally considered as mature T cells. It is generally termed maturation the process that T cells undergo in the thymus where they begin to express the TCR and CD4/CD8 co-receptors while undergoing positive and negative selection. The same applies to the sentence “This suggests that there is either a selective recruitment of mature T cells into the tumor domain or, ...”.

13. Figure 4h. The units of the y-axis should be defined.

14. Table S2: authors claim that the reduction in pS6 was mainly observed in cancer cells, but a graph like the one of Figure 4i for vimentin is lacking.

15. Supplementary Figure 6b-c: They appear blurry in the provided version.

Reviewer #4 (Remarks to the Author): Expert in lung cancer mouse models

This elegant study by Van Maldegem et al. combines the development of both new CyTOF-based techniques and image analysis methods to characterize the composition of the tumor microenvironment (TME). The authors apply this new approach to characterize the effects of KRAS G12C inhibition on the numbers, differentiation and localization of immune cells in the TME. The authors observe major differences in two macrophage populations that have distinct localization and change in response to MRTX1257. This is an important study that provides new insight into the effects of MRTX1257 on the TME. However, the authors should address the following points:

1. Several studies have shown the presence of Tertiary Lymphoid Structures (TLS) in syngenic tumor models. In lung cancer patients the presence of TLS correlates with better prognosis. TLS contain B and T cell zones in addition to DCs and other cell types that can be identified using the panels established by the authors. Are there TLS structures basally in any of the transplant models used? If so, what is their cellular composition, cell interactions and does MRTX1257 treatment impact TLS's.
2. The characterization of CD4 T cells is minimal, the authors should characterize them further. For example, T regs play a very important role in suppressing anti-tumor T cell responses. The authors should assess T reg numbers, localization and interactions with other cell populations, particularly other T cells basally and in response to MRTX1257. This will provide important insight into whether T regs play a functional role in response to MRTX1257 treatment. It's possible that despite the fact that T cell numbers increase upon MRTX1257 treatment their function is suppressed by T regs.
3. Several recent studies have identified populations of macrophages in the lung that have both distinct localization and function (e.g. PMID: 30872492, 32220976). The authors should comment about the potential relationship of the observed macrophage populations to what is currently known about macrophages in the lung.
4. Although this may be beyond the scope of this study, the cellular niche of type2 PD-L1 macrophages associating with DCs and T cells in MRTX1257 treated tumors suggest that anti-PD1 treatment may synergize with MRTX1257. The authors could test this experimentally.

Response to Reviewers' Comments

Nature Communications NCOMMS-21-08959 van Maldegem et al.

7 July 2021

(Reviewers' original comments in non-italic, authors' responses in blue/italic.)

Reviewer #1 (Expert in multi-plex imaging)

Van Maldegem and colleagues report on an imaging mass cytometry panel and analytical tools for mouse tissue and, in parallel, have studied spatial immunophenotypical alterations related to the use of MRTX1257 on a KRAS-mutated mouse tumor model. The reporting of a ready-to-use IMC panel for mouse tissue together with an interesting cell segmentation approach are the main contributions of the paper. The most relevant immunological changes associated to the treatment with MRTX1257 have already been reported elsewhere.

Main comments:

1. I think the presentation of the results should be simplified. There is currently a mixture between methodological and biological questions throughout the paper that do not facilitate its reading. I think the authors should, first, solely focus on the methodological developments implemented by them and then move to their application in the chosen model. In line with this, I think the Results section can be reduced considerably.

We have moved the figure 1c with the example tumours of the two treatments to figure 3a. This leaves the first part (figures 1&2) focussed on the methodology and keeps all the results related to biology and response to therapy together.

2. The choices made to visualize the data in Figure 3 are not the most useful ones. For instance, in Figure 3a, it is very difficult to evaluate the contribution of each marker for the definition of a particular subset. Re-scaling marker expression per subset might be helpful?

We have significantly revised this figure. It now includes the images that were originally in figure 1c, as explained in our reply to the previous point. As the reviewer recommends, we have scaled the heatmap on a row basis to emphasise the contribution of the markers to each cluster. We have moved the tSNE data to supplemental figure S4 and instead focussed on the changes in domain distribution for the different cell types.

3. I think the data has been over-clustered when considering the amount of markers that have been employed and when looking at Figure 3b. Some clusters should be merged in case they do not represent biologically

We have merged clusters as the reviewer recommended and separated out a Regulatory T cell cluster. This leaves us with 13 cell types that takes away the need to separately regard the 35 clusters and the metaclusters that we used previously. It is these 13 clusters that have been taken forward in the rest of the figures. The original 35 clusters are still detailed in an (unscaled) heatmap in Supplementary Figure S4 to share this raw data with the readers.

4. The authors mention PD-1 as being expressed mainly by CD8+ T cells, but in Supplementary figure 5e, PD-1 expression seems to be mainly derived from CD4 T cells?

Yes, the reviewer is right. For conciseness reasons we only focussed on the CD8 T cells, but for completeness we have now included data on PD-1 expression by CD4 and Regulatory T cells in Figure 5.

Minor comments:

1. Is CD44 really a bonafide stem-cell marker when considering that all tumor cells in Figure 1C are staining for it? Isn't CD103 also a marker for intraepithelial/tumor-resident T cells? MHC-II, general activation marker?

CD44 is indeed a marker for stemness/stem cells, but also for activation of many immune cells, and often upregulated in tumours undergoing EMT. This tumour has many characteristics of EMT (e.g. lack of expression of epithelial marker EpCAM for example). CD103 is a marker for human and mouse Tissue resident memory T cells, but we do not see this marked expression in our data. The only population that clearly expresses CD103 is a small subset of Regulatory T cells. And yes, besides a lineage marker for DCs MHC-II is also an activation marker for macrophages, which we observe in the macrophage characterisation of Figure 4. We have updated Figure 1b to include this information.

2. Is CT26 IMC data available? It would be interesting to see whether, despite the lack of response to the inhibitor, immune cell infiltration was nevertheless promoted.

Unfortunately, we do not have data on CT26 to show immune cell infiltration with IMC. In the preclinical work from Amgen (Canon et al, Nature 2019) and Mirati (Briere, Mol Cancer Ther. 2021) it was shown the CT26 cell line did respond to the inhibitor, led to increased lymphocyte infiltration as assessed by flow cytometry and synergised with anti-PD1.

3. ""Treatment of 3LL NRAS tumours established (...) although most tumours did not actually reduce in size (Supplementary Figure 2)."" The lack of response in CT26 should also be included in the figure as this sentence mentions a comparison with control mice.

See point 3, we do not have CT26 data. Clearly our wording has generated confusion about whether we have data on CT26. To avoid this confusion, we have taken out the reference to CT26 from the results section, and instead included some text on the comparison to CT26 in the discussion.

4. ""Vehicle treated tumors usually showed a tissue architecture..."" I think one of the most striking features in addition to the accumulation of macrophages in the surrounding of the tumor is the infiltration by Ly6G-positive cells. This should have been highlighted?

Yes, we agree that the infiltration and aggregation of neutrophils in the vehicle treated tumours is striking, especially as we see that these patches of neutrophils largely disappear in the MRTX treated tumours. We included additional comments regarding the neutrophils in the main text description of Figure 3a.

5. ""Macrophage phenotypes changed"" – this expression is too ambiguous as the authors probably cannot distinguish a change in phenotype from a de novo infiltration by other subsets.

In our view the slingshot analysis supported the idea of a differentiation gradient in the macrophage population; we have modified the text to be less ambiguous about this.

6. Figure 1C – it's very difficult to distinguish CD11c and CD68, I suggest that CD11c is remove or that another panel is included.

We have removed CD11c from this panel (now figure 3a).

Reviewer #2 (Expert in segmentation methods)

The authors present an imaging mass cytometry (IMC) workflow to analyze mouse tumors, suitable for preclinical research. The workflow contains a good preselection of 27 antibodies targeting typical markers and can be extended using 12 custom antibodies. The authors image lung tumors from 3 mice of vehicle and 3 mice of treatment group and analyze the data. They compare 4 spatial spatial cell segmentation algorithms and evaluate their properties. Cells are classified into types using clustering with manually adjusted gates. Based on the cell classification, the tissue is segmented into domains (e.g. tumor vs. healthy tissue) and cell populations are assessed. Furthermore, cell-cell distances, e.g. between tumor and CD8+ cells are quantified and some example marker expression profiles are analyzed. Overall, the imaging technology and analysis software is convincingly showcased using a relevant mouse model. The software (and data) is published with the paper and the solution based in

Nextflow appears to be highly reproducible. The provided tool seems to be of potentially high value for preclinical research and is claimed to be the first of its kind. I think that this work is a valuable contribution but several issues need to be addressed:

1. Introduction

At the end of the introduction, I am missing a statement about the aim of the study, giving a rational and overview for the following sections.

We have added a section to the end of the introduction outlining the work to be presented.

2. Results

Section "An antibody panel for imaging mass cytometry on mouse tissues"

The functionality was nicely shown in a spleen follicle.

Why are there 12 free slots if 27 are taken and 40 are available?

The 40 marker panel was published in the Ijsselsteijn paper, where they included channels for Cadmium, Indium and Platinum isotopes. These metals are not (yet) recommended or validated by Fluidigm for use on IMC and therefore we did not account for those in the free slots. Text has been adjusted to be less specific about the number of free slots.

3. Section "Targeted inhibition of KRAS G12C in a preclinical model of lung cancer"

While the paper is well-written in general, with good figures, I had some trouble understanding the experimental design. It should be made clear at the beginning that there are two groups (vehicle-treated and MRTX1257-treated), with 3 mice each. The sentence "A total of twelve tumours were selected for IMC, six tumours from three mice of each treatment group" is too unclear and I had to read the whole paper several times to figure it out.

We have added a section to set out the experimental design preceding the section highlighted by the reviewer.

4. Is "Orthotopic transplants" the right term for tumor induction by tumor cell injection?

We have reworded it to be less ambiguous.

5. The mention of the CT26 tumor is confusing here, as it conveys the impression that you had an experimental CT26 group you are referring to.

We have reworded and moved the discussion on CT26. Unfortunately, we do not have data on CT26 to show immune cell infiltration with IMC. In the preclinical work from Amgen (Canon et al, Nature 2019) and Mirati (Briere, Mol Cancer Ther. 2021) it was shown the CT26 cell line did respond to the inhibitor, led to increased lymphocyte infiltration as assessed by flow cytometry and synergised with anti-PD1.

6. The discussion of F4/80+ and CD206+ macrophages and CD4+ and CD8+ T cells is a bit confusing here, because there is a much more thorough discussion later.

This discussion is now part of Figure 3. We have moved the figure 1c with the example tumours of the two treatments to figure 3a. This leaves the first part (figures 1&2) focussed on the methodology and keeps all the results related to biology and response to therapy together.

7. Section "Optimisation of single cell segmentation using cell type specific settings"

Since this seems to be an important innovation or part of the tool, this should be mentioned in the outlook at the end of the introduction.

A section has been added to the end of the introduction to describe the outlook of the paper, introducing the workflow including the cell type specific segmentation, as well as further details on the work to be discussed.

8. Please briefly mention the scope or purpose of the libraries CellProfiler and Ilastik.

We have added two sentences to describe in short what the packages provided to the segmentation.

9. Section "IMC single cell analysis reveals dramatic remodelling of TME"

This is a interesting section but it is too long. Please break it down, maybe into clustering, domain analysis, cell-cell-interactions, macrophages, lymphocytes, and marker-expressions (vimentin).

We have reorganised the results section and broken down into separate sections. To provide more discussion of biological insights, we have made separate figures for the topics of macrophages, T cells and vimentin, including additional analyses for each topic.

10. What I lack most in the results (and figures) is a statistical analysis comparing the two groups, e.g. that the mean distance to of tumor to CD8+ cells decreased significantly in the treatment group compared to the untreated group. This means that for each mouse one score is generated, we have three scores per group (3 mice per group) and a test (e.g. t-test) showing a significant difference for the 3 vs 3 values. Each mouse is a sample, not each tumor. Such (simple) statistical comparisons between groups are widespread in preclinical research and therefore would be interesting for the intended audience which is supposed to apply the proposed tool. Such an analysis would also give an impression of the inter-group variance of such scores.

The reviewer is right that statistics should have been included, but we were unsure how to approach this. Reducing down the data to just 3 mice in each treatment group ignores much of the variation that exists between ROIs of an individual mouse. Especially when we look at infiltration of certain cell types into the tumour we see that there is a wide range of "responses" within one mouse. We have called in the help of a statistician (G. Kelly) who, after careful consideration of all variables in the data, has put together a statistical analysis using a linear mixed effects model. This accounts for fixed effects of domain, cell type and treatment, whilst allowing for a per-mouse and ROI-within-mouse variation of distribution of cells between cell types.

11. Discussion:

Please mention some the differences between human and murine cell images. Are mouse cells/nuclei smaller?

We are not aware of general differences in cell size between mouse and human, but the need for optimising segmentation came mostly from the use of frozen tissue sections and the densely packed nature of the tumours. The discussion was modified to better explain this.

12. When saying "Seeing the tumour as a whole..." please mention that you still analyze a single slice only.

This section has been modified accordingly.

13. Methods:

Please provide some details about the μ CT scan, e.g. voltage, power, exposure, projection size and count, voxel size.

Methods section on μ CT scanning has been modified to contain more detail.

14. Please explain: "treatment groups were randomised with stratification by tumour number and size"

We rephrased this sentence.

15. Figures:

Figure 2: a) please mention sequential segmentation. Maybe put panel b at the beginning.

This figure was adapted to contain one large panel describing all segmentation strategies.

Reviewer #3 (Expert in tumour microenvironment)

Van Maldegem, Valand et al. developed a useful automated pipeline for the analysis of imaging mass cytometry (IMC) data obtained from frozen mouse tissue sections. Availability of such pipelines is highly valuable for the unbiased structural evaluation of the tumor microenvironment (TME). The authors refined existing tools previously developed by Bodenmiller and others (largely developed for FFPE sections of human tumors) and improved the analysis of the spatial distribution of cells in the TME of frozen mouse tumors. However, it appears that the novelty of the approach is only incremental, as clearly acknowledged by the authors. In fact, the pipeline is largely based on algorithms developed by Bodenmiller and others (ImcPluginsCP and IMCtools for segmentation; Rphenograph and Louvain community detection algorithm; tSNE, uMAP for clustering; neighbourhood for spatial analysis and Slingshot pseudotime trajectory analysis; etc.). This said, the availability of the new workflow amenable to the analysis of frozen mouse tissues is certainly valuable and would be of broad interest.

Regarding the biological findings of the manuscript – the characterization of mouse lung tumors treated with a KRAS-G12C inhibitor – the results are descriptive and do not seem to add much to what could be obtained using alternative methods. Indeed, the main results are consistent with previous work (Canon et al. Nature 2019; Hallin et al. Cancer Discovery 2020; Adachi et al Clin Cancer Res 2020; Singh et al Cancer Cell 2009).

In summary, this is a useful technical report that will advance the application of IMC to frozen mouse tissues. However, at this stage of development, application of the refined method did not unravel new biology of KRAS-G12C inhibition.

Specific comments.

1. Figure 3a: could authors explain the reasons for excluding some markers in the clustering? In fact, adding Foxp3 would have been important to identify regulatory T cells, which have very distinct biology compared to other CD4 T cells (re bulk of CD4 06 cluster).

We have tried to cluster with all markers but found that this does not always lead to the most logical results, e.g. separation of cells by phenotype rather than cell type. We figured clustering by lineage markers would give us the cell types, after which the phenotypes could be studied. We found that even when Foxp3 was included as marker for clustering, Regulatory T cells were included with other CD4 T cells in a single cluster. Therefore we decided to manually separate these cells by thresholding.

2. Figure 3a and 4a: authors claim that cluster 11 corresponds to type 1 macrophages. However, it is not clear how they distinguished them from dendritic cells, as both display low F4/80 expression. It would be good to list the exact markers that were used for the definition of each cluster in a supplementary item (the current description is not exhaustive enough).

Cluster 11 was annotated as macrophages based on the high CD68 expression, another pan-macrophage marker. We have generated a supplementary table S3 that now has listed the criteria based on which each cluster was assigned, including thresholds where gating was used to separate clusters. In combination with Figure S4a, a heatmap depicting all clusters and all markers (unscaled), we hope that this will provide sufficient insight in how the clusters were annotated.

3. Figure 4d. Authors perform a Slingshot pseudotime trajectory analysis to assess the differentiation of macrophages. Such algorithms were created to delineate possible differentiation trajectories from high dimensional datasets like scRNAseq. However, in the case of this study only a few parameters were considered for such analysis. In fact, only 6 markers are relevant for macrophages (authors should clarify if only specific macrophage markers were used for the analysis or all markers were included). Therefore, the suitability of such algorithms and their advantages over standard

characterization of different macrophage populations and expression of individual markers (CD68, MHCII, CD206, CD11c, PDL1, CD86) are apparently questionable. In addition, as expression of each marker was based on a segmentation that is not perfect, differences in expression could derive from technical errors that are not compensated by analysis of hundreds of variables (as it happens when analysing scRNAseq data). What are the results of performing the same Slingshot pseudotime trajectory analysis by separating treated and untreated tumors?

We used 8 markers for the generation of the uMAP and the slingshot pseudotime trajectory analysis (F4/80, CD68, CD11c, MHCII, CD206, CD11c, PDL1, CD86). The expression of these markers is shown in the uMAPs that we have now given a place in the main figure, rather than supplementary. We have added these details to the methods section as well.

A previous publication from the Bodenmiller group also performed pseudotime analysis, on pancreatic islet cells using 12 markers (PMID: 30713109). When we ran the Slingshot on the MRTX treated data separately, the plot looked exactly like the one shown in the original figure. The slingshot analysis to us supported the notion that the gradients of marker expression in the type 2 macrophage after MRTX1257 treatment reflected a continuous differentiation path. But we agree that it can't be excluded that the differences in expression could be derived from imperfect segmentation and that the low number of markers is perhaps not optimal for such analysis. As the pseudotime inference was not meant to be a big statement, we have decided to take it out from the manuscript.

4. Figure 4e: Why are there different numbers of data points (presumably samples) for type1 and type2 macrophages in the neighbourhood analysis? Using the same scale for the analysis of type 1 and type 2 macrophages would make the graph easier to interpret.

Not all cell types occur in all images, which occurs most frequently for epithelium.

Calculating a LOG2FC will in such instance divide by zero and therefore generate an "Infinity", which cannot be plotted.

We have fixed the scales of the y-axis to make the comparison between type 1 and type 2 macrophages easier.

5. Figure 4e: It may be interesting to assess the neighbourhood of CD8 T, CD4 T, T regs and B cells separately as it is known that macrophages can play distinct functions on different lymphocyte subtypes. Also, it would be interesting to perform such neighbourhood analysis on the tumor edge vs tumor core to reinforce the statement that "CD68+ macrophages are found often in close proximity to T cells and CD103+ DCs, particularly at the edge of the tumour (Supplemental Figure 5b). This is also true of the minority of intra-tumoural CD68+ macrophages, which maintain neighbourhoods with T cells and dendritic cells". Is it possible to assess whether the neighbourhood enrichment is significantly different in type1 vs type 2 macrophages? According to the graphs, it seems that log2FC enrichment values of type 1 and type 2 macrophages vs DCs are similar.

As a result of restructuring the cell types we have now got CD4, CD8 and regulatory T cells as separate populations in the neighbourhood analysis. As suggested, we have separated the neighbourhood analysis for the macrophages in figure 4 in the different domains. This has revealed that the interaction between macrophages type 1 and dendritic cells is very conserved also in the tumour domain, but for T cells this is less so. Our additional analysis looking at the neighbours of T cells in figure 5d also suggests that the interaction between type 1 macrophages and T cells is more indirect, via close interactions with dendritic cells. We have adapted the results section on the macrophages to better reflect these interactions. We have compared the neighbourhood enrichment plots from the two types of macrophages using the linear mixed-effects modelling. Plots to represent these statistics are included in Supplemental figure 5.

6. Figure 1B. CD4 is also a marker of regulatory T cells in addition to Th cells. CD44 is also a differentiation marker for T cells.

Figure 1b has been updated to include this information.

7. Figure 3a-c. What markers were used to identify normal lung (non-tumor associated) cell clusters? It would be good to list the exact markers that were used for the definition of each cluster.

The “normal lung” clusters were annotated based on the lack of expression of any of the immune, fibroblast, tumour or endothelium markers, and localisation in normal tissue. But reconsidering this, we have now assigned them as “unclassified”, accepting that our panel is not exhaustive enough to ensure that this is not still a mixture of cell types. In answer to point 3 we have described the new table S3 to list all markers that defined the cluster annotations.

Minor points

8. Supplementary Figure 4a-b. What do the grey bars represent? If the bar colouring is correct, it looks like there are no fibroblasts or macrophages in vehicle-treated mice and no neutrophils in MRTX-treated mice.

Unfortunately, some data-loss had occurred in the generation of the PDF during submission. There was not supposed to be any grey on those figures. This has been corrected.

9. Supplementary Figure 4d: Is there an error in the image of 03_vehicle? There is no segmentation observed but a big purple ball.

Again this was due to data loss during file compression. Extra care has been taken to avoid this in this resubmission.

10. Supplementary Figure 5d: It would be interesting to plot CD4 and CD8 T cells with different colours to see if they show distinct distributions.

This figure (now Sup Figure S6a) has been updated to depict CD4, CD8 and regulatory T cells in different colours.

11. In the introduction the authors state “it has become apparent that tumours are infiltrated with a diverse spectrum of immune cells, often with different phenotypes from their normally homeostatic counterparts 1, 2”. Ref 1 is not adequate as does not assess tumor-infiltrating cells.

Indeed this reference does not regard tumour infiltrating cells. It was included as landmark paper on CyTOF but is not in the right context here. It has been replaced with more relevant references.

12. Figure legends should define the sample size if it varies among panels.

We have mentioned the subsampling that was done for the visualisation of the tSNEs in the legend of Supplemental figure 4. Otherwise analysis was always done on the whole dataset.

12. In the sentence that states “greater expression of the maturation and exhaustion marker PD-1”, it would be more appropriate to say “activation and exhaustion marker PD1”, as naïve T cells are generally considered as mature T cells. It is generally termed maturation the process that T cells undergo in the thymus where they begin to express the TCR and CD4/CD8 co-receptors while undergoing positive and negative selection. The same applies to the sentence “This suggests that there is either a selective recruitment of mature T cells into the tumor domain or, ...”.

That is indeed a mistake, which has been corrected.

13. Figure 4h. The units of the y-axis should be defined.

The units were in pixels, which has been added to the axis-title.

14. Table S2: authors claim that the reduction in pS6 was mainly observed in cancer cells, but a graph like the one of Figure 4i for vimentin is lacking.

While the reduction in pS6 was limited to tumour cells and dendritic cells, when we reanalysed this data per ROI and per mouse and found no significant differences in pS6 between the treatment groups. We have therefore decided to omit this statement.

15. Supplementary Figure 6b-c: They appear blurry in the provided version.

This was again due to file compression, see points 8 and 9

Reviewer #4 (Expert in lung cancer mouse models)

This elegant study by Van Maldegem et al. combines the development of both new CyTOF-based techniques and image analysis methods to characterize the composition of the tumor microenvironment (TME). The authors apply this new approach to characterize the effects of KRAS G12C inhibition on the numbers, differentiation and localization of immune cells in the TME. The authors observe major differences in two macrophage populations that have distinct localization and change in response to MRTX1257. This is an important study that provides new insight into the effects of MRTX1257 on the TME. However, the authors should address the following points:

1. Several studies have shown the presence of Tertiary Lymphoid Structures (TLS) in syngenic tumor models. In lung cancer patients the presence of TLS correlates with better prognosis. TLS contain B and T cell zones in addition to DCs and other cell types that can be identified using the panels established by the authors. Are there TLS structures basally in any of the transplant models used? If so, what is their cellular composition, cell interactions and does MRTX1257 treatment impact TLS's.

This lung cancer model based on the 3LL Δ NRAS cell line does not associate with mature tertiary lymphoid structures. Occasionally we observe some very small lymphoid aggregates where B cells, T cells and DCs colocalise, but in a disorganised structure and usually associated with larger blood vessels. But mostly B cells are found diffusely distributed in the normal tissue surrounding the tumours and that doesn't change with MRTX1257 treatment.

2. The characterization of CD4 T cells is minimal, the authors should characterize them further. For example, T regs play a very important role in suppressing anti-tumor T cell responses. The authors should assess T reg numbers, localization and interactions with other cell populations, particularly other T cells basally and in response to MRTX1257. This will provide important insight into whether T regs play a functional role in response to MRTX1257 treatment. It's possible that despite the fact that T cell numbers increase upon MRTX1257 treatment their function is suppressed by T regs.

We have now separated the T cells into CD4, CD8 and regulatory T cells for all analyses from figure 3 onwards and dedicated a separate figure to the characterisation of the T cells. Figure 6S now shows the distribution of all three subsets in the tissue, as well as a quantification of these T cell types and their presence in the tumour. This shows that regulatory T cells do not accumulate as much as the CD8 T cells in response to MRTX1257, except in ROI MRTX_07 where the tumour is flooded with all T cell types. The neighbourhood analysis also offers insight into these cell types separately, and this has been discussed in the Discussion section.

3. Several recent studies have identified populations of macrophages in the lung that have both distinct localization and function (e.g. PMID: 30872492, 32220976). The authors should comment about the potential relationship of the observed macrophage populations to what is currently known about macrophages in the lung.

We thank the reviewer for suggesting these studies that describe distinct subsets of macrophages in the lung. In addition to these papers, we have also looked at three other papers that describe macrophage populations in lung tumours (PMID: 30979687, 28475900 and Casanova-Acebes et al, Nature 2021)). We have included several references and a more detailed discussion regarding the macrophage subtypes in both the results and discussion sections.

4. Although this may be beyond the scope of this study, the cellular niche of type2 PD-L1 macrophages associating with DCs and T cells in MRTX1257 treated tumors suggest that anti-PD1 treatment may synergize with MRTX1257. The authors could test this experimentally.

This is indeed something that we are actively looking into, but these are not trivial experiments and as the reviewer already indicates, we believe that to be beyond the scope of this manuscript.

Reviewer #1 (Remarks to the Author):

The authors have done an excellent job in revising their manuscript. It reads much better than previously, especially the connection and flow between their methodological innovation and biological interrogations.

My only comment is that it is not very easy to interpret figure 4D. I suggest that the authors attempt to simplify it. Maybe a heatmap displaying only interactions that are significantly increased? Any format that allows the reader to immediately pinpoint which interactions are increased in each group.

Reviewer #3 (Remarks to the Author):

The authors have clarified the technical points raised during peer-review and have included additional data and analyses.

A couple of further suggestions:

1. Figure 1B: CD44 is a differentiation marker of T cells. In current Fig. 1B it appears that CD44 is a tumor activation marker, instead of T cell differentiation marker.

2. While co-existence between type 1 and type 2 macrophages with DCs and CD4 T cells seems comparable in Figure 4d and Supplementary figure 5b, greater differences are seen when looking at the enrichment of type 1 and type 2 macrophages in the neighborhood of DCs and CD4 T cells (Supplementary figure 5a). The authors may consider moving those graphs from Supplementary figure 5a to main Figure 4, and moving to the supplementary figures the neighbourhood analysis based on tumor site (current Figure 4d). Supplementary Figure 5b should be mentioned together with Figure 4d and Supplementary Figure 5a in the results section.

Rebuttal to additional suggestions reviewers Sept 2021

REVIEWERS' COMMENTS

Reviewer #1 (Remarks to the Author):

The authors have done an excellent job in revising their manuscript. It reads much better than previously, especially the connection and flow between their methodological innovation and biological interrogations.

My only comment is that it is not very easy to interpret figure 4D. I suggest that the authors attempt to simplify it. Maybe a heatmap displaying only interactions that are significantly increased? Any format that allows the reader to immediately pinpoint which interactions are increased in each group.

Reviewer 1 and 3 both comment on the neighbourhood plots of Fig 4d and Supplementary Fig 5, that were revised in the resubmission following suggestions of reviewer 2.

We prefer not to use a heatmap as reviewer 1 suggests, as that would discard too much of the relevant information. Furthermore, that would not allow for any of the comparisons that were originally requested by reviewer 2. We do appreciate that the reader may be helped with some visual aids to improve interpretability. Therefore we have added error bars, that give an easier interpretation of the data spread, and we have highlighted the information that we want to focus attention to with coloured boxes.

Reviewer #3 (Remarks to the Author):

The authors have clarified the technical points raised during peer-review and have included additional data and analyses.

A couple of further suggestions:

1. Figure 1B: CD44 is a differentiation marker of T cells. In current Fig. 1B it appears that CD44 is a tumor activation marker, instead of T cell differentiation marker.

We have added CD44 as differentiation marker on T cells and Myeloid cells to figure 1b.

2. While co-existence between type 1 and type 2 macrophages with DCs and CD4 T cells seems comparable in Figure 4d and Supplementary figure 5b, greater differences are seen when looking at the enrichment of type 1 and type 2 macrophages in the neighborhood of DCs and CD4 T cells (Supplementary figure 5a). The authors may consider moving those graphs from Supplementary figure 5a to main Figure 4, and moving to the supplementary figures the neighbourhood analysis based on tumor site (current Figure 4d). Supplementary Figure 5b should be mentioned together with Figure 4d and Supplementary Figure 5a in the results section.

We have followed the advice of reviewer 3 to move the per-domain plots that were requested by reviewer 2 to the Supplementary figure, and include plots for the macrophages as well as the plots for the CD4 T cells, dendritic cells and fibroblasts in Fig 4d and 4f to show the reciprocal neighbourhood enrichment.